# Compound electron acceleration at planetary foreshocks

Xiaofei Shi [1] ✉, Anton Artemyev [1], Vassilis Angelopoulos [1], Terry Liu [1] & Lynn B. Wilson III [2]

Shock waves, the interface of supersonic and subsonic plasma flows, are the primary region for charged particle acceleration in multiple space plasma systems, including Earth's bow shock, which is readily accessible for in-situ measurements. Spacecraft frequently observe relativistic electron populations within this region, characterized by energy levels surpassing those of solar wind electrons by a factor of 10,000 or more. However, mechanisms of such strong acceleration remain elusive. Here we use observations of electrons with energies up to 200 kiloelectron volts and a data-constrained model to reproduce the observed power-law electron spectrum and demonstrate that the acceleration by more than 4 orders of magnitude is a compound process including a complex, multi-step interaction between more commonly known mechanisms and resonant scattering by several distinct plasma wave modes. The proposed model of electron acceleration addresses a decades-long issue of the generation of energetic (and relativistic) electrons at planetary plasma shocks. This work may further guide numerical simulations of even more effective electron acceleration in astrophysical shocks.

In collisionless space plasmas, shock waves heat and energize charged particles[1–3]. Astrophysical shocks are believed to generate some of the most energetic particles in the universe[4–6]. A long-lasting mystery in shock acceleration is how to accelerate background thermal particles up to superthermal or even mildly relativistic energies (so-called Fermi's injection problem). In-situ spacecraft measurements at interplanetary shocks[7] and at bow shocks of inner[8] and outer planets[6] of the heliosphere are the most natural way to test and explore this particle energization. Such measurements are very copious and detailed at Earth's bow shock and its foreshock, the region upstream of the Earth's bow shock which contains many solitary, large-scale transient structures[9]. This region has been found to host acceleration of electrons by more than four orders of magnitude, from solar wind energies of ≤ 10 electron volts (eV) to near-relativistic energies of hundreds of keV[8,10,11]. This acceleration is very effective, given the relatively limited scale-size of the foreshock, and has been an unsolved issue for decades. Investigating the mechanisms responsible for the formation of such energetic electrons upstream of the collisionless shock will provide unique information for models and theories of shock acceleration in various space plasma systems[12].

Classic shock-drift acceleration (SDA) alone is insufficient for accelerating solar wind electrons to hundreds of keV without effective electron trapping around the bow shock[13,14]. Stochastic shock drift acceleration (SSDA)[11] overcomes this limitation by assuming pitch-angle (angle between velocity and magnetic field direction) scattering of electrons at turbulent wave field sites on either side of the shock. This allows electrons that bounce between those sites to spend sufficient time near the shock to be shock-drift accelerated to high energies[11,12]. SSDA's efficiency depends on the effectiveness of pitch-angle scattering by electron resonant interactions with electromagnetic and electrostatic waves[15,16].

There is no single wave mode that can pitch-angle scatter electrons efficiently over the wide energy range from 10 eV to hundreds of keV. However, multiple wave modes exist in the bow shock and foreshock, such as: electrostatic waves consisting of a mixture of ion-acoustic waves[16,17], ion and electron phase space holes[18]; electromagnetic high-frequency whistler-mode waves[19,20]; low-frequency whistler-mode

[1]Department of Earth, Planetary, and Space Sciences, University of California, Los Angeles, California, USA. [2]NASA Goddard Space Flight Center, Heliophysics Science Division, Greenbelt, MD, USA. ✉e-mail: sxf1698@g.ucla.edu

(magnetosonic) waves[21,22]; and ultra-low frequency magnetic field perturbations[23,24]. Each wave mode can resonate with electrons in a specific (often quite narrow) energy range, but acting together, these modes may cover the entire energy range of interest.

When there is a large Sun-Earth component to the magnetic field in the solar wind, occasional discontinuities transported by the latter interact with the foreshock and cause foreshock transients. These are sites of large magnetic field compressions that can adiabatically reflect or locally heat electrons[10]. Moreover, these transients often form new shock waves ahead of the bow shock, contributing to electron SDA upstream of the parent shock[25]. A subset of electrons in the foreshock environment is subject to scattering and acceleration by many or all these processes, as depicted in Fig. 1, amounting to a compound, aggregate acceleration. Such acceleration is difficult to study except by using appropriate modeling with realistic assumptions, well-guided by observations.

Here, we show that the combination of electron resonant scattering by different wave modes, electron adiabatic reflection from large-amplitude foreshock transients, and SDA can collectively account for the formation of observed electron fluxes up to and above 200 keV, consistent with observations. Toward this goal, we employ observations of Magnetospheric Multiscale (MMS), Time History of Events and Macroscale Interactions during Substorms (THEMIS), theoretical models of wave-particle resonant interaction with electrostatic and whistler-mode waves, and a probabilistic approach which allows for rapid evolution of electron trajectories in prescribed magnetic fields and wave fields. The MMS data are used to derive the statistical properties of the wave fields with high spatial resolution around transients, whereas the THEMIS data are used to inform us of the typical spatial structure of the foreshock environment.

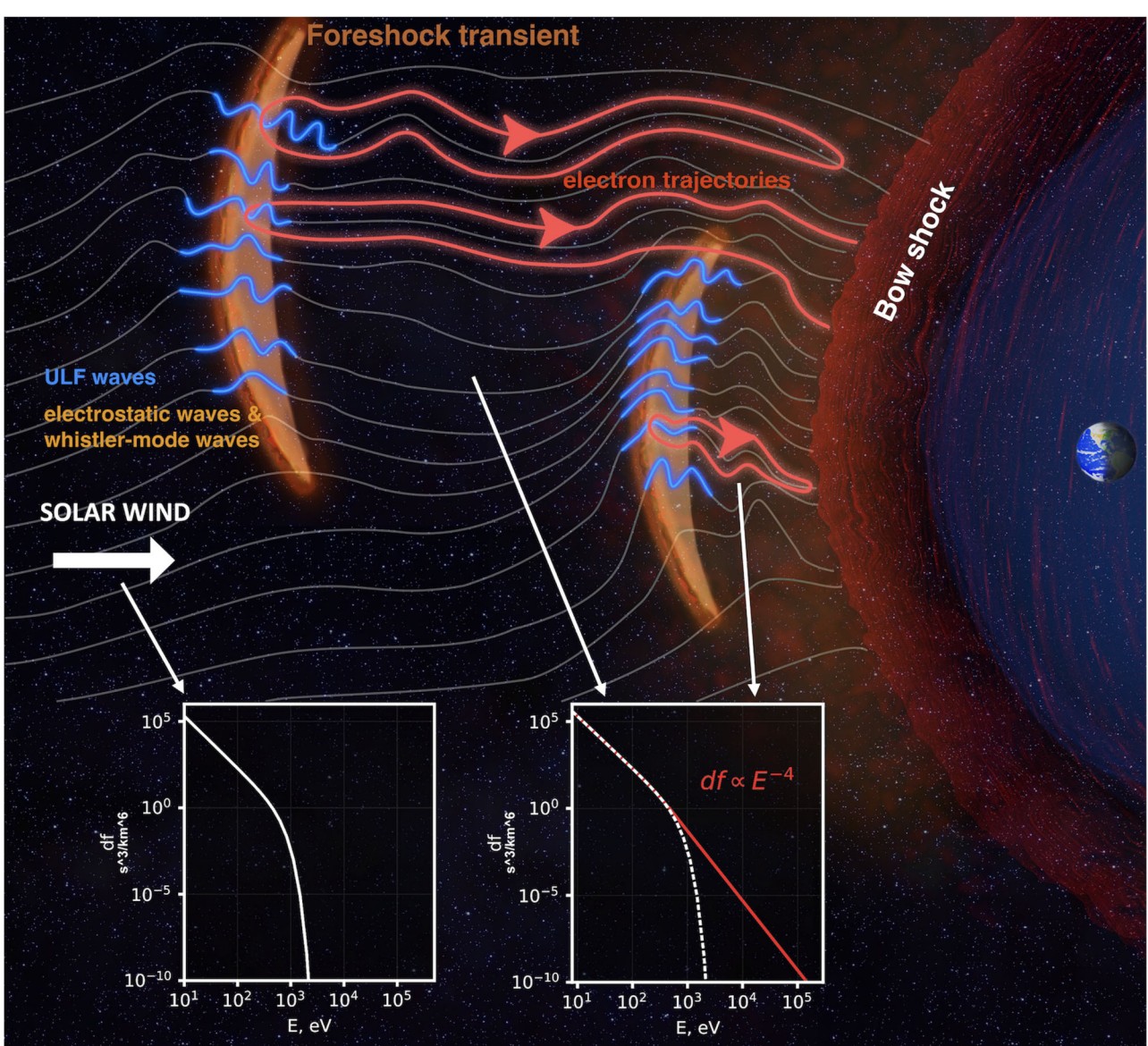

**Fig. 1 | Schematic of electron interaction with the bow shock and high-amplitude magnetic field transients in the foreshock region.** An electron can traverse or bounce at these strong field regions. Between successive bounces, electrons are scattered in pitch angle by electrostatic waves, high-frequency whistler-mode waves, and magnetosonic waves. During the bounce motion, electrons can be accelerated through adiabatic acceleration, including shock drift acceleration (SDA) at the bow shock, SDA at the boundary of the foreshock transient (to distinguish this acceleration mechanism from SDA at the bow shock, we call it Fermi acceleration), and betatron acceleration due to compression and magnitude increase of the magnetic field in the core region between the bow shock and the transient boundary. The original electron phase space density (df) in the solar wind decreases quickly as energy (E) increases above about 100 eV. After acceleration, the electron phase space density distribution has a power-law tail ($df \propto E^{-4}$) up to hundreds of keV, as shown in the red line.

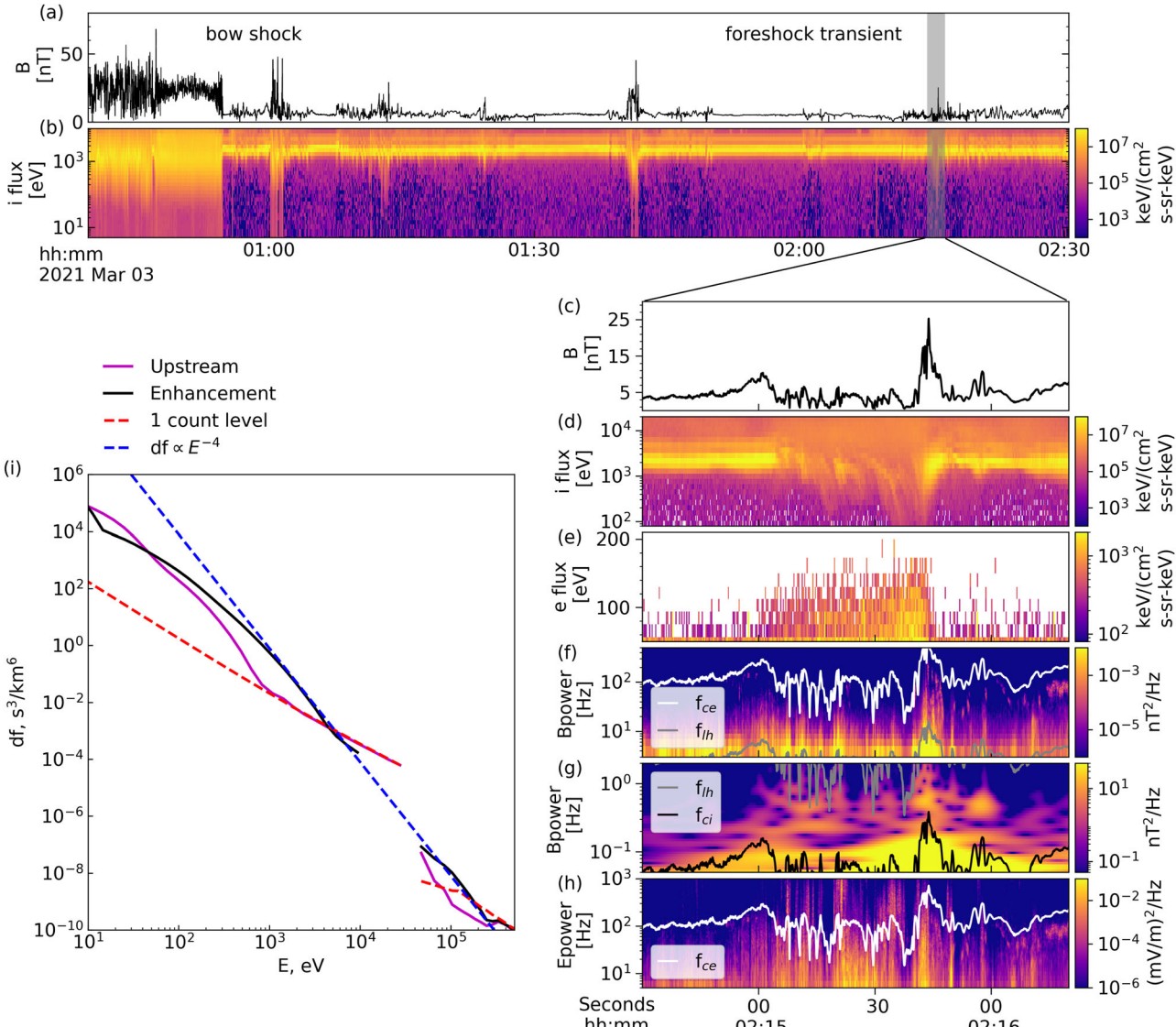

**Fig. 2 | Observations of flux enhancement of tens to hundreds of keV electrons at a foreshock transient.** Between 00:30 and 02:30 UT, MMS crossed Earth's bow shock and foreshock, where multiple foreshock transients were detected. Panels (**a** and **b**) show magnetic field strength and ion energy spectra, respectively. Panels (**c**–**h**) zoom into a subset of the above interval during a foreshock transient event. From top to bottom, these panels show: (**c**) the magnetic field strength, (**d**) the ion energy spectrum, (**e**) the electron energy spectrum for 50–200 keV electrons indicating the presence of relativistic electrons up to 150 keV, (**f**, **g**) magnetic field power spectra for high and low frequencies, respectively; (**h**) the high-frequency electric field power spectrum; electrostatic turbulence observed around the electron cyclotron frequency (white lines). The white lines in (**f**, **h**) are electron cyclotron frequency ($f_{ce}$), the gray lines in (**f**, **g**) are low hybrid frequency ($f_{lh}$), and the black line in (**g**) is ion cyclotron frequency ($f_{ci}$). Panel (**i**) illustrates the observed electron phase space density (df) at the upstream region outside transients (magenta lines) and at the foreshock transient of panels (**c**–**h**) (blue lines). Reliable measurements are limited to data above the 1 count level (dashed red line). Notably, the electron df during the enhancement adheres to a power-law behavior, with df proportional to $E^{-4}$ (dashed blue line). Source data are provided as a Source Data file.

## Results

To justify our main theoretical assumptions and motivate our choice of model parameters, we present in Fig. 2 MMS observations of foreshock transients, exhibiting significant electron fluxes around the 200 keV range (See "Methods", subsection MMS observation for additional information on spacecraft instrumentation and data processing). MMS crosses the quasi-perpendicular bow shock around 00:50 UT. The shock has a normal [0.99, −0.03, 0.10] in geocentric solar ecliptic (GSE) coordinates at an angle of 80°±5° to the interplanetary magnetic field. The shock-normal velocity is 650 km/s. The bow shock crossing is evident as a clear transition from thermalized ion energy spectra and a strong, highly fluctuating magnetic field intensity downstream, to narrow spectra and a weakly fluctuating field upstream. The foreshock region (in the upstream) is replete with energetic ions (≥10 keV) coincident with transient

magnetic field enhancements. These are typical observations of foreshock transients[9,26]. We zoom into one of them at ≈02:15UT: there is a classic transient configuration with two magnetic field boundaries (peaks at 02:15 and 02:16UT in Panel (c)) and a core region characterized by weak and fluctuating magnetic field, reflected ions, and strongly scattered solar wind beam (Panel (d)). We focus on the energetic electrons filling the core region (Panel (e)) up to 200 keV. The electron phase space density energy-spectrum (Panel (i)) shows a power law $E^{-4}$ tail ($E$ denotes electron energy) between energies about 1 keV and 200 keV, in good agreement with previously reported energetic electron events at foreshock transients by THEMIS[8,10]. To gain such high energies, the solar wind electrons must be able to interact multiple times with the bow shock. Therefore, there should be some mechanisms providing solar wind electron trapping between the bow shock and its foreshock transients.

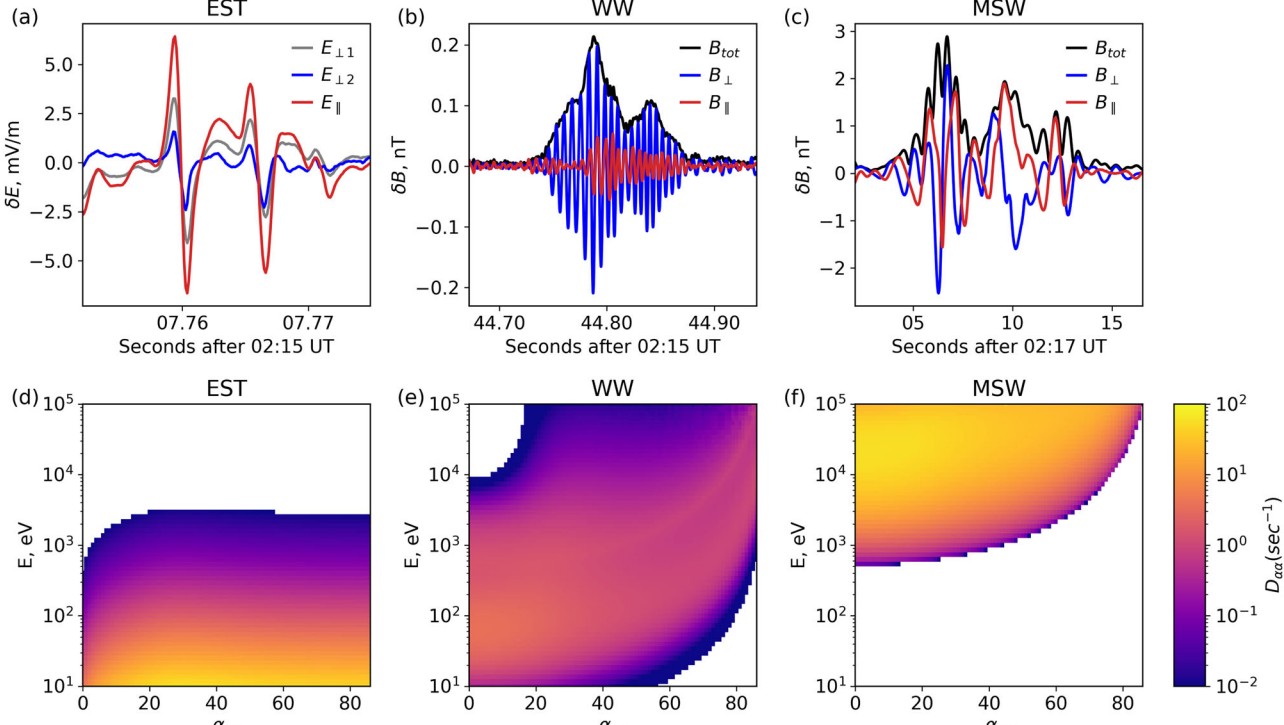

**Fig. 3 | Representative waveforms of three types of wave modes typically observed around the compressional boundary of foreshock transients, selected from times of the example shown in Fig. 2, along with pitch-angle scattering rates associated with typical wave power for such wave modes. a** Large-amplitude electrostatic (EST) wave seen in electric field measurements ($\delta E$). **b** High-frequency whistler-mode (WW) wavepacket, with predominant magnetic field perturbations in the perpendicular direction, suggesting propagation along the background magnetic field. **c** Wavepackets the magnetic field for magnetosonic (MSW) waves. Observation times (UT) are listed in the horizontal axis. Panels (**d–f**) depict diffusion coefficients for the three types of waves averaged over background magnetic field conditions representative of the large-scale magnetic field perturbation. Source data are provided as a Source Data file.

Trapping can be accomplished by adiabatic reflection or electron pitch-angle scattering as follows: First, the foreshock region is filled with transients with large magnetic field fluctuations[23,25]. The compressional nature of these fluctuations may allow them to adiabatically reflect electrons, trapping them between the bow shock and the ensemble of transients within the foreshock (such electrons will be bouncing). Second, the foreshock magnetic boundaries host intense electrostatic turbulence[16–18] (Fig. 2h), high-frequency whistler mode waves[19,20] (Fig. 2f), and low-frequency magnetosonic waves[21,22] (Fig. 2g). These wave modes provide effective pitch-angle scattering for electrons covering a wide energy range. Figure 3a–c shows typical waveforms of these wave modes. Electrostatic turbulence is dominated by ion-acoustic solitary waves, ion holes, and electron holes, all having spatial scales about Debey length and propagation velocities within the range between ion to electron thermal speeds, which are [$10^{-3}$, $10^{-4}$] times the speed of light[16–18]. High-frequency electromagnetic whistler mode waves have spatial scales (wavelength) about the electron inertial length and propagation speed about $10^{-2}$ of the speed of light[19,20]. Low-frequency magnetosonic waves have spatial scales (wavelength) about the ion inertial length and propagate at a speed of [$10^{-3}$, $10^{-4}$] of the speed of light[22]. The efficiency (rate) of electron pitch-angle scattering by these waves is given by the pitch-angle diffusion rate ($D_{\alpha\alpha}$, in $rad^2/s$, also see Methods, subsection Electron resonant scattering rates). The mechanics for calculating the diffusion rates for these waves are well-established[18,27,28] for a homogeneous or weakly inhomogeneous magnetic field. However, the foreshock region is also filled with large amplitude magnetic field fluctuations, which provide strong inhomogeneity along electron bounce trajectories. Thus, we averaged the standard rates of pitch-angle diffusion over an ensemble of observed magnetic field fluctuations, using THEMIS statistics (see "Methods", subsection Spatial scale of the electron acceleration

region) to establish the spatial scales of these fluctuations. Figure 3d–f shows such averaged rates: electrostatic turbulence (EST) with frequencies above the electron cyclotron frequency ($f_{ce}$) mostly scatters <1 keV electrons of large and intermediate pitch-angles, high-frequency whistler mode waves (WW) with frequencies between low-hybrid frequency ($f_{lh}$) and $f_{ce}$ mostly scatter <10 keV electrons with the resonant energy increasing with pitch-angle, and magnetosonic waves (MSW) with frequency between ion cyclotron frequency and $f_{lh}$ mostly scatter >10 keV field-aligned electrons. Figure 3 confirms that the three wave modes cover five orders of magnitude in energy, from the solar wind electron energy of about 10 eV to near the maximum observed energy of energetic electrons of about 100 keV.

The acceleration mechanism includes electron SDA at the bow shock (determined by the shock speed; see Methods, subsection MMS observation), and Fermi (SDA at the foreshock transient boundary) and betatron (determined by magnetic field increase within the core region) acceleration upstream of it (see "Methods" subsection Numerical simulation approach) due to the foreshock transient motion[29]. Note that although high-frequency whistler mode waves also contribute to electron acceleration in our model[20], the main role of all three wave modes and compressional fluctuations is to trap electrons near the bow shock, allowing them to experience multiple SDA and adiabatic (Fermi and betatron) acceleration.

To reproduce the observed electron spectrum, we simulate electron dynamics affected by a combination of electron scattering and acceleration (the numerical methods are expounded in Methods, subsection Numerical simulation approach). We start the simulation with the solar wind electron distribution (gray color), where most of the electrons have energies below 100 eV (although there is a finite population up to 1 keV), and aim to resolve the question of electron acceleration up to 100 − 200 keV (see Fig. 4). The model considers that the electrons are

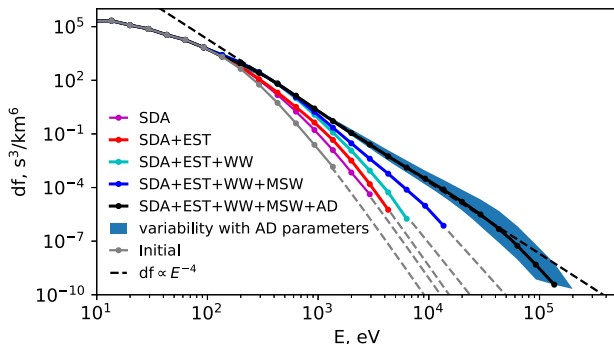

**Fig. 4 | Results of our modeling of the compound electron acceleration process, successively incorporating various effects considered in this paper.** Formation of the observed $E^{-4}$ energy spectrum necessitates the inclusion of all the effects considered in this study. Gray solid line: initial electron distribution; Magenta line: model results considering only the SDA acceleration at the bow shock; Red, green, and blue lines: progressive addition of pitch-angle scattering by EST, WW, and MSW waves, respectively; Black line: adding the Fermi acceleration (SDA at foreshock transient boundary) and betatron acceleration in the core region between the bow shock and foreshock transient to all the other effects. Only when all the effects discussed are compounded does the result adhere to df $\propto E^{-4}$, consistent with observations. The blue-shaded region indicates the variability in the results when different adiabatic parameters are chosen (e.g., the mean value of shock normal speed is varied in the range of 800–1200 km/s; see Methods, subsection MMS observations for details). The dashed gray lines fit the high energy portion of the spectra, shown for clarity. Source data are provided as a Source Data file.

bouncing in the magnetic bottle formed between the bow shock and foreshock transient boundary. When we include only SDA acceleration at the bow shock, most electrons escape the foreshock region after about 7 reflections from the shock, and the maximum acceleration does not exceed $1 - 3$ keV (magenta curve). If EST scattering is added, electrons may experience up to about 12 reflections from the shock and gain $3 - 5$ keV (red curve). The inclusion of electron WW scattering and acceleration increases the number of electron reflections from the shock further, up to about 24, with the electron energy gain reaching $5 - 7$ keV (green curve). Note that WW can also change electrons' energy (see details in "Method", subsection Numerical simulation approach). Scattering of more than a few keV electrons requires the inclusion of MSW scattering, which increases the number of electron reflections from the bow shock to about 40, whereas the electron energy gain reaches $10 - 20$ keV (blue curve). Therefore, a combination of SDA at the bow shock, and resonant scattering by three different wave modes can provide acceleration of <100 eV solar wind electrons to around 20 keV. The next effects to be included are the Fermi (SDA at the foreshock transient boundary) and betatron adiabatic acceleration upstream of the bow shock. The boundary of the foreshock transient often forms its shock wave propagating upstream ahead of the bow shock with a velocity comparable to that of the bow shock. Since the foreshock transient is moving towards the bow shock, electrons reflected by the transient structure experience SDA at the transient boundary (we refer to it as Fermi acceleration)[30]. As the foreshock transient moves (collapses) onto the bow shock, it compresses the magnetic field in the core region, increasing by a factor of approximately 3 (see ref. [10]). This effect should provide electron betatron heating in the core region between the bow shock and foreshock transient[10]. Figure 4 shows that when both Fermi and betatron adiabatic (AD) acceleration are also included in the simulation, the electron spectrum reaches about 200 keV and attains a power law falloff of about $E^{-4}$ (black curve). The variability of the model output spectrum due to uncertainties in the bow shock speed determination is depicted by the blue-shaded region (see "Methods", subsection Numerical simulation approach for details).

The good agreement of the model results of Fig. 4 with observations in Fig. 2i validates the proposed scenario of solar wind electron

acceleration in the foreshock to $100 - 200$ keV. Such acceleration transforms a small subset of the initial <1 keV solar wind electron distribution (comprising a core population below 100 eV and an exponential energy tail) into a power law distribution with an $E^{-4}$ falloff, a power law in the 1–200 keV energy range. Importantly, efficient electron acceleration from solar wind energies, around $10 - 100$ eV, up to near-relativistic energies, about 200 keV, cannot be otherwise explained by a single mechanism of electron scattering. Rather, multiple wave modes and adiabatic acceleration by foreshock transient compound the energy gain arising from SDA at the bow shock, allowing electrons of progressively higher energies to continue to be scattered upstream and have the opportunity to be further accelerated. The proposed compound acceleration mechanism successfully reproduces the electron acceleration by a factor of 10, 000 within the compact region of the foreshock and thus provides a quantitative solution to the problem of collisionless shock acceleration that has remained unresolved for decades. This mechanism reveals the key role played by multiple wave modes (ES, EE, MSW) in trapping electrons (via pitch-angle scattering) within the foreshock region and providing stable conditions for electron energization. The same plasma kinetics (wave-particle resonant interactions) may resolve electron acceleration at other heliospheric[6] and astrophysical[4,5] shock regimes with appropriate parameter scaling. The generalization and scaling of such a model require detailed numerical simulations of wave activities[11]. This is because the crucial wave characteristics for our model cannot be obtained through in-situ observations in these space plasma systems. Therefore, the proposed and verified acceleration mechanism is expected to deepen our understanding of particle acceleration at shocks physics and may be important in particle acceleration and the generation of cosmic rays at other astrophysical settings shocks throughout the cosmos.

## Methods
### MMS observations
The example from Fig. 2 shows foreshock observations of energetic electrons by the Magnetospheric Multiscale (MMS) mission[31]. The high time resolution of plasma and field measurements by MMS's are particularly well suited for studies of the plasma and wave properties inside foreshock transients. We use burst mode magnetic field data from the fluxgate magnetometer (FGM)[32] and the search coil magnetometer (SCM)[33] at a rate of 128 S/s and 8192 S/s, respectively. FGM provides information about the background magnetic field and the low-frequency (magnetosonic) whistler-mode waves, whereas the SCM dataset provides the main characteristics of high-frequency whistler-mode waves. The electron and ion energy spectra are obtained from combined measurements of the fast plasma investigation (FPI)[34] instrument, covering energies < 25 keV and energetic electron spectra are obtained from the Fly's Eye Energetic Particle Spectrometer (FEEPS)[35], covering energies $50 - 650$ keV. FGM magnetic field measurements and FPI plasma flow vector measurements are combined in the coplanarity method[36] to estimate the bow shock normal, whereas the shock normal speed (in the spacecraft frame) is calculated using the mass conservation law[36]. For the bow shock crossing at 00:55UT in Fig. 2, we obtain the normal angle of $80° \pm 5°$ and the shock normal speed ($v_{bs}$) of 650 km/s at the time interval 00:54:30–00:55:00 UT (downstream time interval: 00:54:30–00:54:45 UT, upstream time interval: 00:54:45–00:55:00 UT).

### Spatial scale of the electron acceleration region
To estimate the spatial scale of the foreshock region filled by the transient structures and hosting electron acceleration, we use the statistics of observations by THEMIS that comprises five (2008-2009) and three (2010–2023) satellites[37]. We focus on the 2010–2023 period, when THEMIS A, D, and E can near-simultaneously traverse the dayside region, maintaining a spatial separation ranging from hundreds to

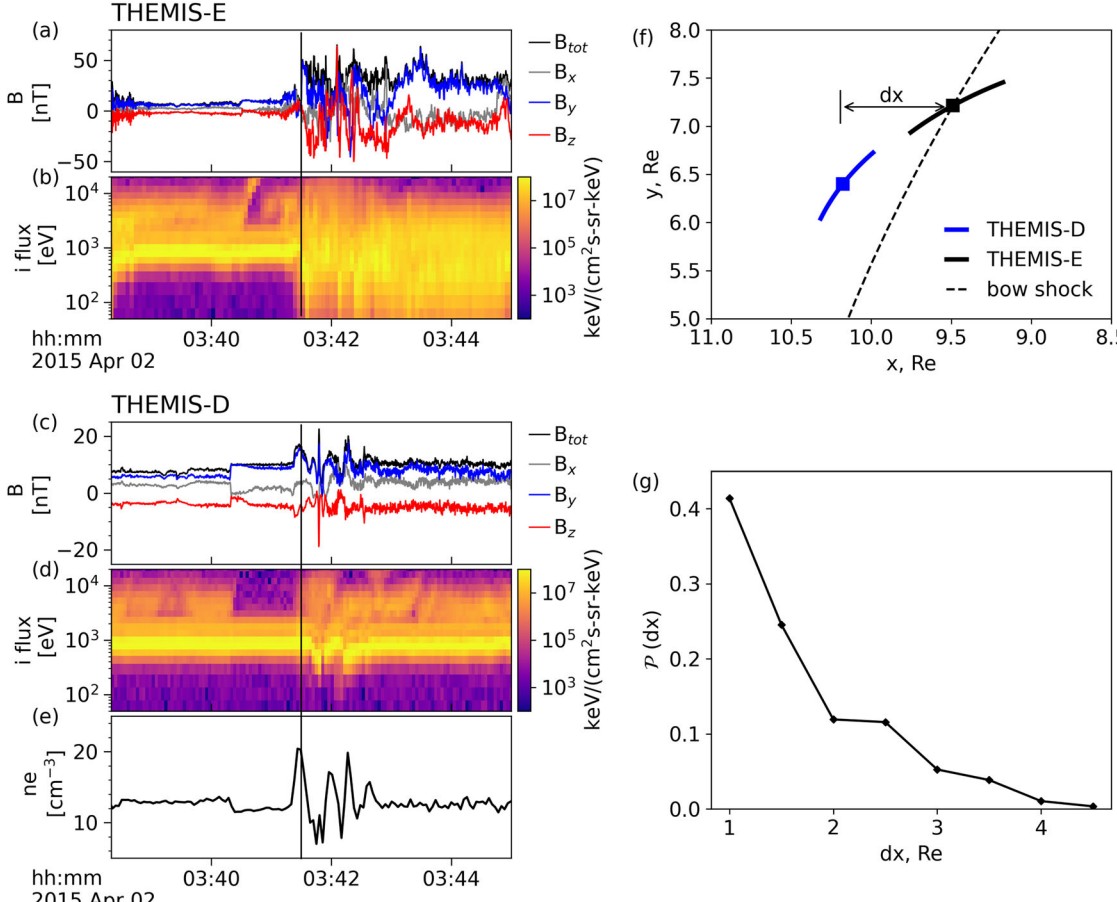

**Fig. 5 | Typical multi-satellite observation of a foreshock transient's plasma environment detected by THEMIS. a**, **b** THEMIS-E crossing of the bow shock, denoted by the vertical black line. **c**–**e** concurrent THEMIS-D (nearby THEMIS-E) observation of the foreshock transient (denoted by the vertical black line), identified by its magnetic field enhancement and electron density perturbation in the upstream region; (**f**) locations of THEMIS-D and THEMIS-E in the GSE coordinate system, with the dashed black line indicating the position of the bow shock. We have found ~100 similar events, involving one THEMIS satellite crossing the bow shock and another being in the upstream region observing foreshock transient perturbations; (**g**) statistical distribution of the distances between the bow shock and transient structure, offering insights into the spatial scales of the foreshock acceleration region. Source data are provided as a Source Data file.

thousands of km (several Earth radii). We use the routinely (always) available magnetic field and plasma moment data provided by the THEMIS fluxgate magnetometer (FGM)[38] and ion electrostatic analyzers (iESA)[39], respectively. We assemble a database of events when one of the THEMIS spacecraft crosses the bow shock, and another one observes the foreshock transient, similar to the example in Fig. 5a–f. Using such events, we estimate the distance between the upstream foreshock transient structures and the bow shock. This distance serves as the spatial scale of the electron acceleration region in our model, where electrons bounce between the bow shock and the boundary of foreshock transients. To compile statistics, we utilize the THEMIS dataset from 2010–2023, select events akin to the one described above, and impose the constraint that the distance between the two satellites in the GSE-Y direction should be less than 2.5 Earth radii since we are mostly interested in the spatial scale along solar wind flow. The distribution of the spatial scales in the database is depicted in Fig. 5g. Most of the foreshock transients are observed within 4 Earth radii (about 25000 km) upstream of the Earth's bow shock, in agreement with previously published estimates[40]. This spatial scale is used in our model of electron dynamics.

**Electron resonant scattering rates**

Our model of electron resonant scattering due to the wave-particle interactions in the foreshock region includes electrostatic turbulence, high-frequency whistler-mode waves between the lower-hybrid $f_{lh}$

and the electron cyclotron $f_{ce}$ frequencies, and low-frequency whistler-mode magnetosonic waves, a continuation of whistler-mode below $f_{lh}$. For each of these modes, we use the theoretical model of quasi-linear scattering rate (pitch-angle diffusion coefficient), $D_{\alpha\alpha}$. Intense electrostatic turbulence around the Earth's bow shock consists of different nonlinear waves and packets of intense ion acoustic waves[16,18]. The scattering rate for such turbulence has been derived and verified in[16,41]:

$$D_{\alpha\alpha} = \int \frac{D_{\alpha\alpha}^{(X)}\mathcal{P}(X)\ell(X)dX}{\int \ell(X')\mathcal{P}(X')dX'}, \quad X = \left(v_\phi, \ell, \theta, \mathcal{E}_w\right) \quad (1)$$

$$D_{\alpha\alpha}^{(X)} = \frac{\mathcal{E}_w^2\Omega_e^2}{4\sqrt{2\pi}N_eE\Omega_{pe}} \frac{(\ell/\lambda_D)^3}{\sin^2\alpha} \frac{\left(v_\phi\cos\alpha - v\cos\theta\right)^2}{|v_\phi - v\cos\alpha\cos\theta|^3} \sum_{n=1}^{n=\infty} n^2 J_n^2(\rho_n)e^{-\xi_n}$$

$$(2)$$

$$\xi_n = \frac{n^2\Omega_e^2(\ell/\lambda_D)^2}{\Omega_{pe}^2\left(v_\phi - v\cos\alpha\cos\theta\right)^2}, \quad \rho_n = \frac{nv\sin\alpha\sin\theta}{v_\phi - v\cos\alpha\cos\theta} \quad (3)$$

where $\ell$ is the spatial scale of nonlinear waves, $\lambda_D$ is the Debye length evaluated with the background electron density $N_e$ and temperature, $v_\phi$ and $\theta$ are the phase speed and normal angle (relative to the

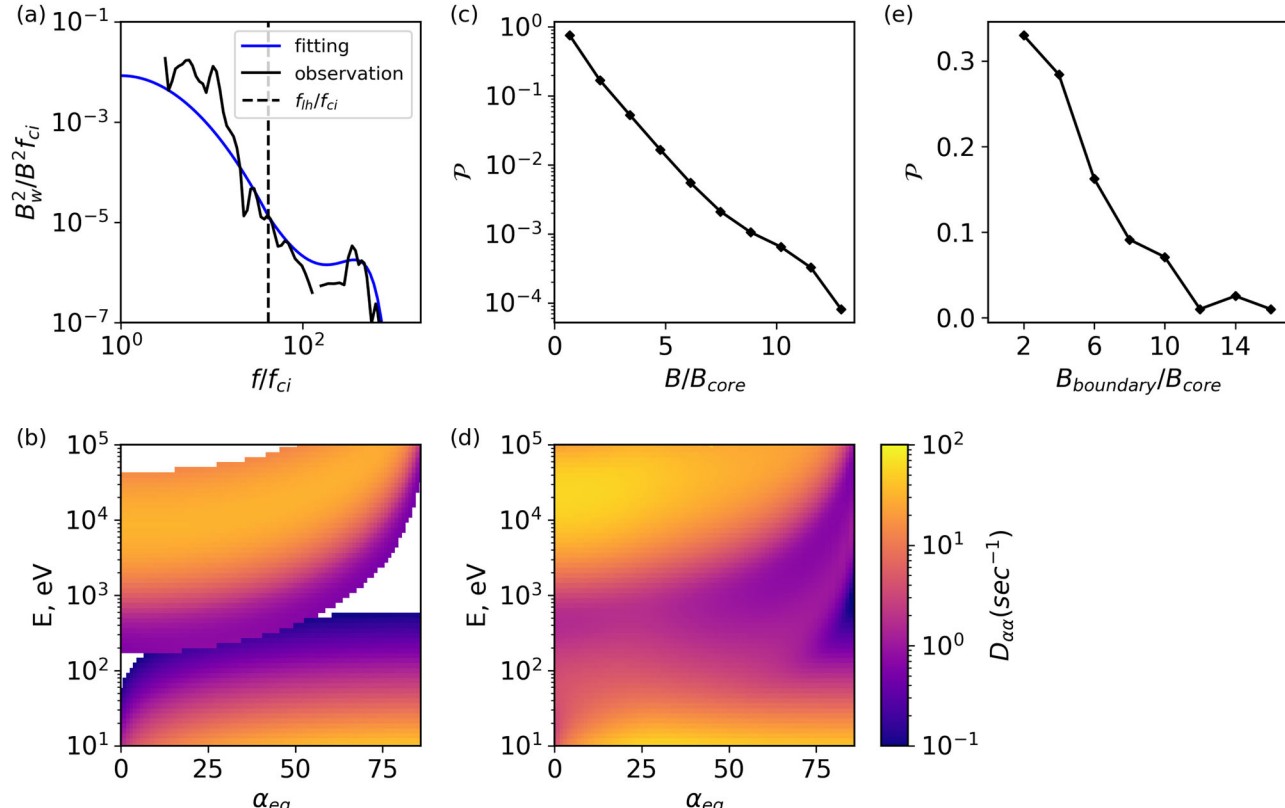

**Fig. 6 | Derivation of simulation parameters for waves. a** observed magnetic field power spectrum of magnetosonic waves and high-frequency whistler-mode waves (black lines) and a fit to the observations (blue line). The vertical dashed line indicates the low hybrid frequency; (**b**) combined diffusion coefficient for the three types of waves calculated at the minimum magnetic field (inside the core); (**c**) distribution of magnetic field perturbations within foreshock transient structures; (**d**) combined diffusion coefficient averaged over the background magnetic field perturbations; (**e**) distribution of the ratio $B_{boundary}/B_{core}$. Source data are provided as a Source Data file.

background magnetic field) of nonlinear waves, $\mathcal{E}_w$ is the wave electric field amplitude, $\Omega_{ce}$ and $\Omega_{pe}$ are the ambient electron gyrofrequency and plasma frequency, respectively, $n$ is the number of cyclotron resonance, $J_n(\rho_n)$ is the Bessel function, $v$ is the velocity of electrons with the energy $E$. The first integral is the averaging of the diffusion rate $D_{\alpha\alpha}^{(X)}$ over the probability distribution $\mathcal{P}$ of wave characteristics X (see details of $\mathcal{P}(X)$ in ref. 16).

The diffusion rate for the whistler-mode waves has been derived[27], and we use the approximation for field-aligned waves resonating with electrons, including relativistic corrections (this relativistic correction becomes important for ≥100 keV energies)[28]:

$$D_{\alpha\alpha} = \frac{\int D_{\alpha\alpha}^{(X)} \mathcal{P}(X) dX}{\int \mathcal{P}(X) dX}, \quad X = (\omega, \mathcal{B}_w) \quad (4)$$

$$D_{\alpha\alpha}^{(X)} = \frac{\pi \mathcal{B}_w^2 \Omega_{ce}}{2 B_0^2 \gamma^2} \frac{(v - v_\phi \cos\alpha)}{|v\cos\alpha - v_g|} \left|\frac{v_g}{v}\right| F(\omega) \quad (5)$$

where $\omega$ is the wave frequency, $\gamma$ is the relativistic factor, $\mathcal{B}_w/B_0$ is the ratio of wave magnetic field amplitude and the background field, $v_g = \partial\omega/\partial k$ is the wave group velocity derived from the cold plasma dispersion relation[42], and $F(\omega)$ stands for the power spectrum of the waves. The normalization in the first line provides $\int_{\omega_-}^{\omega_+} F(\omega) d\omega = 1$, where $\omega_\pm$ denotes the lower and upper limits of the frequency range, respectively. Notably, magnetosonic waves and high-frequency whistler-mode waves represent the same wave mode (whistler-mode) but with different frequencies: for magnetosonic waves, the frequency range extends from the proton cyclotron frequency to the low-hybrid

frequency, whereas for high-frequency whistler-mode waves the frequency range spans from the low-hybrid frequency to the electron cyclotron frequency. The diffusion coefficient for both types of waves can be estimated using Equations (4) and (5). We conducted separate fittings for the power spectrum, $F(\omega)$, of magnetosonic waves and whistler-mode waves, as illustrated in Fig. 6a.

Each type of wave has a specific resonance energy range: electrostatic waves predominantly interact with electrons below 100 eV, high-frequency whistler-mode waves primarily influence electrons in the range of 10 eV to 1 keV, and magnetosonic waves exert substantial effects on electrons with energies exceeding 1 keV. Figure 6b shows the combined diffusion coefficient of electrostatic waves, high-frequency whistler waves, and magnetosonic waves. The diffusion coefficients are evaluated for the $B_0 = B_{core}$, where $B_{core}$ is the magnetic field magnitude in the core region. However, the foreshock transient region exhibits high amplitude variations of the magnetic field, as depicted in Fig. 6c (where $\mathcal{P}$ represents the probability distribution of background field fluctuations relative to $B_{core}$). Under such intense magnetic field fluctuations, the wave-particle resonance conditions can strongly vary, widening the energy range of electrons scattered by each of the three wave modes[43]. To account for this effect, we calculated the diffusion coefficient averaged over the background magnetic field fluctuations.

$$\langle D_{\alpha\alpha}(\alpha, E) \rangle = \frac{\int_0^{B_{max}} D_{\alpha\alpha}(\alpha, E, B_0) \mathcal{P}(B_0) dB_0}{\int_0^{B_{max}} \mathcal{P}(B_0) dB_0} \quad (6)$$

Such averaged diffusion coefficient $\langle D_{\alpha\alpha} \rangle$ for the three types of waves is shown in Fig. 3 of the main text, while the combined result is presented

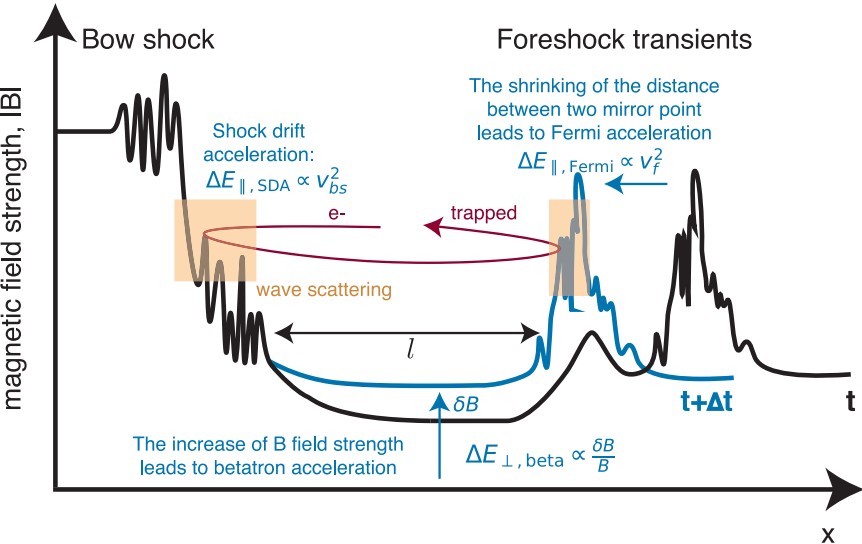

**Fig. 7 | A schematic view of the main elements of the electron acceleration model.** Electrons are bouncing between two magnetic walls (mirrors) formed by the bow shock and the foreshock transient boundary (the red curve shows the electron bouncing trajectory, the black curve shows the magnetic strength profile shaping the effective potential for electron trapping, $l$ is the distance between the bow shock and the foreshock transient). With bow shock normal velocity $v_{bs}$, electrons experience SDA with the energy gain $\Delta E_{\parallel,\text{DSA}} \propto v_{bs}^2$. The motion of the foreshock transient boundary with velocity $v_f$ also results in SDA acceleration of electrons, which is termed Fermi acceleration in the paper, $\Delta E_{\parallel,\text{Fermi}} \propto v_f^2$, to separate it from the SDA at the bow shock. When the foreshock transient approaches the bow shock, it compresses plasma in the core region (between the bow shock and foreshock transient), increasing the magnetic field magnitude (blue curve shows the magnetic strength profile with time evolution from $t$ to $t + \Delta t$). Such an increase in the magnetic field magnitude results in beta-tron electron acceleration, $\Delta E_{\perp,\text{beta}} \propto \delta B/B$. When electrons reach the bow shock and the foreshock transient boundary, they experience scattering due to resonant interactions with electrostatic turbulence, high-frequency whistler-mode waves, and magnetosonic waves (orange regions denote the location for wave-particle interactions). In addition, high-frequency whistler waves can also change electron energies during the interaction.

in Fig. 6d. The inclusion of background magnetic field fluctuations leads to broader energy and pitch-angle ranges of finite $\langle D_{\alpha\alpha} \rangle$. The combined effect of these three wave modes, enhanced by background magnetic field fluctuations, facilitates the scattering of electrons across a broad energy spectrum - from 10 eV to 100 keV, encompassing all pitch angles. This overlap provides a continuous path in energy from 10 s of eV to 10;s of keV for electrons to be scattered by these waves. The most intense of these compressional fluctuations, the foreshock transient boundaries, contribute as well by adiabatically reflecting electrons thus increasing the probability of electron trapping.

## Numerical simulation approach

The numerical simulation of electron scattering and acceleration is based on a probabilistic approach[44,45], similar to the Monter-Carlo approach commonly applied to astrophysical shock waves[46,47]. Figure 7 shows a schematic view of the electron dynamics, including all mechanisms of electron scattering and acceleration. An elementary time-step in the model is the electron's bounce period between the bow shock and the foreshock transient, given by $\tau_b \approx l/v_{\parallel}$ where $l$ is a spatial scale of bouncing. For each electron, the simulation starts with a bouncing spatial scale $\ell$ selected from the $dx$ distribution shown in Fig. 5g. As the simulation progresses, $l$ shrinks, affecting the electron bounce time $\tau_b$. If an electron escapes the trapping from a transient structure or the transient structure reaches the bow shock, the electron may start bouncing between the shock and another foreshock transient structure (we assume there can be three transient structures within simulation domain[40], allowing electrons three chances to be reflected by the transient boundary; see discussion below). In this case, $l$ is selected again from the $dx$ distribution. At the end of each bounce period, we check if the electron can continue to the next bounce period. This is determined by whether the electron's pitch angle is large enough to be outside the loss-cone of the bow shock and three foreshock transients. If the pitch angle is too small,

the electron is excluded from the system and replaced by a new electron randomly selected from the initial distribution in the solar wind.

The bow shock magnetic field strength has a spatial gradient increasing by approximately 4 times from the core field. We assume that the shock configuration remains unchanged throughout the entire simulation. For each electron interaction with the foreshock transient boundary, the transient boundary magnetic field $B_{boundary}/B_{core}$ is generated from the distribution function of $B_{boudnary}/B_{core}$ obtained in MMS statistics (see Fig. 6e). Although the number of particles is conserved within the simulation, we assume that the core of solar wind electron distribution remains unchanged (as there is an almost infinite source of solar wind). In Fig. 4, a small population of solar wind electrons with < 100 eV is added to all spectra to make them the same for < 100 eV energy range.

Following the scheme shown in Fig. 7, we describe all mechanisms of electron acceleration and scattering within one bounce period. Interaction with the bow shock results in SDA with the electron parallel energy change, $\Delta E_{\parallel,\text{SDA}} = 2m_e v_{bs}^2$ where $v_{bs}$ is the bow shock speed. Determination of this velocity for a specific shock can have large uncertainties, and this velocity may change within the simulation time. Therefore, to reduce the effect of such uncertainties on our results, we determine $v_{bs}$ for each interaction from a normal Gaussian probability distribution with a mean value of 1000 km/s and a standard deviation of 400 km/s, which corresponds to the shock speed of about 100 km/s (see ref. 48) and shock normal angle above 80° (see ref. 49). We have verified that the changing the mean value to 800 km/s and 1200 km/s only slightly affect the final spectrum of accelerated electrons.

Electrons also experience SDA acceleration when interacting with the foreshock boundary, due to its movement towards the bow shock with velocity $v_f$. To separate this acceleration from SDA at the bow shock, we refer to it as Fermi acceleration (analogous to the acceleration due to the shrinking of $l$ scale)[50]: $\Delta E_{\parallel,\text{Fermi}} = 2m_e v_f^2$. In our

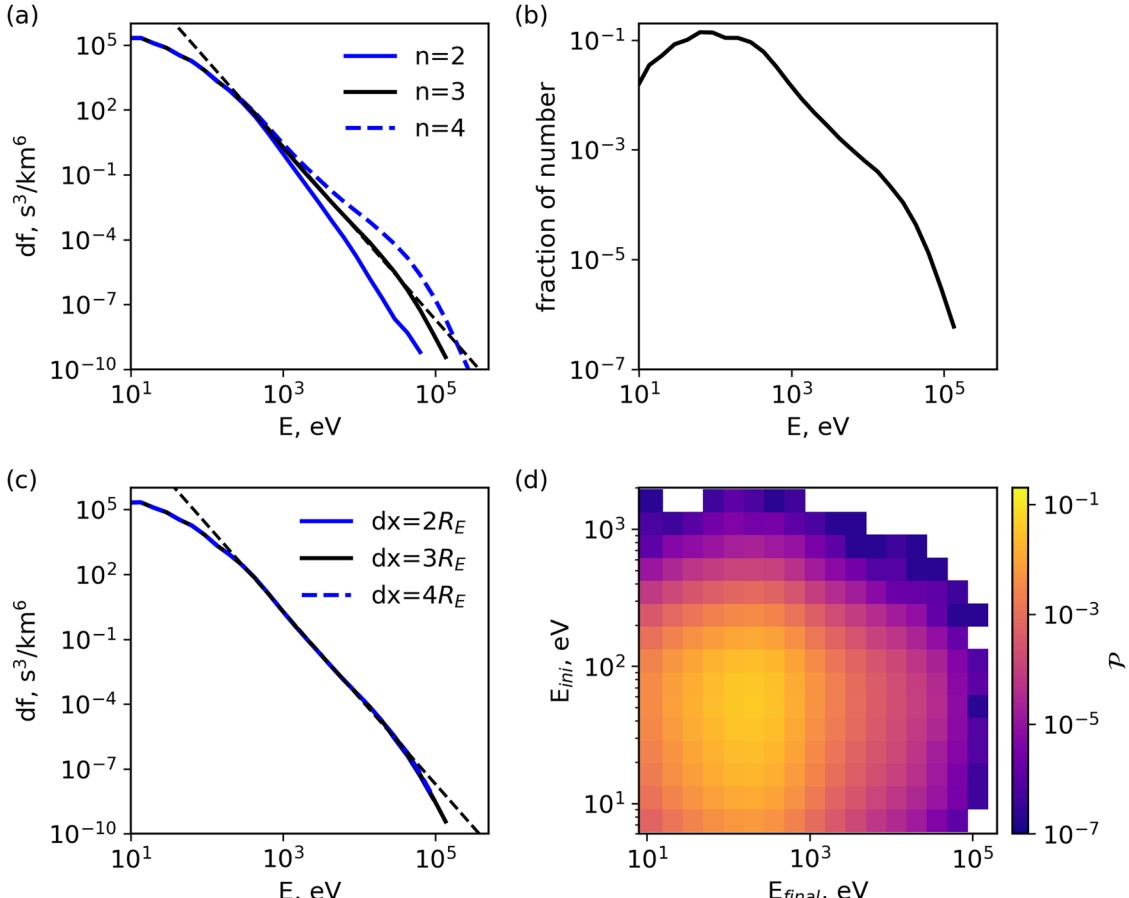

**Fig. 8 | Investigation of the role of multiple foreshock transient crossings in the electron energization. a** effect of the number of foreshock transients ($n_t$) on electron spectrum: as $n_t$ increases, electrons can be accelerated to higher energies; (**b**) normalized number density distribution of the final result of a simulation with $5 \times 10^7$ electrons; (**c**) effect of different distance $dx$: for a given time (here is 10 min), a larger distance results in fewer bounces because of a longer bouncing period; (**d**) probability distribution in the (initial energy, final energy) space, demonstrating that the main source of the accelerated particles is the core of the solar wind (<100 eV). The dashed black lines in (**a**, **c**) are df $\propto E^{-4}$. Source data are provided as a Source Data file.

simulation, $v_f$ is determined using the same normal Gaussian probability distribution as the bow shock speed, $v_{bs}$.

The foreshock transient motion toward the bow shock compresses the plasma in the core region (the region between bow shock and foreshock transient) and results in magnetic field increase[10]. Therefore, electrons experience betatron acceleration with the perpendicular energy increase $\Delta E_{\perp,\text{beta}} = E_\perp((B+\delta B)/B - 1)$, where $\delta B$ is the increase of magnetic field in the core region over a $dt$ time-scale (we consider $\sum \delta B/B = 3$ for the entire simulation period)[10].

In addition to these three acceleration mechanisms, electrons experience pitch-angle scattering as $\Delta \alpha = \sum_i W_i \sqrt{\langle D_{\alpha\alpha}(E,\alpha)\rangle_i}$ where $D_{\alpha\alpha}$ is the diffusion coefficient, $i = EST, WW, MSW$ stands for electrostatic turbulence (EST), high-frequency whistler-mode waves (WW), and magnetosonic waves (MSW), $W_i = \mathcal{N} \cdot \sqrt{dt}$, and $\mathcal{N}(0,1)$ is a random number from the normal Gaussian probability distribution with a zero mean value and unity dispersion[44], $dt = \int_L ds/v_\parallel$ represents the time-scale electrons spend around the foreshock transient boundary (where all wave modes are hosted), with $L = 100$ km denoting the spatial scale of this boundary. The energy change during the interactions of electrons with EST and MSW is negligible due to low frequencies of these waves, and thus the pitch-angle change can be directly recalculated into changes of parallel and perpendicular energy components. Conversely, interactions with high-frequency whistler-mode waves can change electron energy, and we use the relation between $\Delta \alpha$ and energy change[28] to calculate $\Delta E_{\parallel,\text{WW}}$ and $\Delta E_{\perp,\text{WW}}$.

Combining all these effects, we obtain the equations describing electron energy $E$ and pitch-angle $\alpha$ recalculation within one bounce period (between $n$ and $n+1$ bounces):

$$
\begin{aligned}
\alpha_{n+1/2} &= \alpha_n + \sum_{i=EST, WW, MSW} W_i \sqrt{\langle D_{\alpha\alpha}(E_n, \alpha_n)\rangle_i} \\
E_{\parallel, n+1/2} &= E_{\parallel, n} + \Delta E_{\parallel, WW}(E_n, \alpha_n + 1/2) \\
E_{\perp, n+1/2} &= E_{\perp, n} + \Delta E_{\perp, WW}(E_n, \alpha_n + 1/2) \\
E_{\parallel, n+1} &= E_{\parallel, n+1/2} + \Delta E_{\parallel, \text{Fermi}}(E_{\parallel, n+1/2}) + \Delta E_{\parallel, \text{SDA}}(E_{\parallel, n+1/2}) \\
E_{\perp, n+1} &= E_{\perp, n+1/2} + \Delta E_{\perp, \text{beta}}(E_{\perp, n+1/2}) \\
\alpha_{n+1} &= \arctan(E_{\perp, n+1}/E_{\parallel, n+1})
\end{aligned}
\tag{7}
$$

There are two key assumptions regarding the bow shock and foreshock configuration that we need to discuss. First, our simulation spans 10 minutes of real-time, which aligns closely with (or slightly larger than) the convection time required for the foreshock transient to traverse the distance $dx$ shown in Fig. 5g at the typical convection speed[9]. To maintain the trapped electrons within the foreshock for these 10 mins, additional cross-field electron transport, likely caused by wave scattering and the bow shock, would be necessary, though quantifying this transport is beyond our simulation's scope. These electrons can then be trapped by another transient, with up to three such trappings possible in a single bounce period. Thus, the properties of individual transients do not significantly affect the acceleration. Figure 8c shows that the change of the mean distance $dx$ does not

change the final spectrum of acceleration electrons. Second, our acceleration model presumes the existence of the foreshock, which necessitates reflected ion beams and a particular shock normal angle. This assumption prevents us from exploring how acceleration efficiency varies with the shock normal angle. Therefore, further studies using more advanced simulations are needed to quantify the roles of cross-field transport and shock configuration in electron acceleration.

Most of the system parameters for numerical simulations of electron dynamics, scattering, and acceleration are selected according to spacecraft statistical observations. However, the role of a free model parameter, the number of simultaneously existing foreshock transients, requires additional verification. The number of such transients determines the probability of electron adiabatic reflection in the foreshock and thus should affect the electron acceleration efficiency. Figure 8a shows that with other system parameters fixed, the increase/decrease of the number of foreshock transients ($n = 3 \pm 1$) results in a variation of phase space density of $> 10$ keV electrons and maximum energies within $\in [75, 300]$ keV. Therefore, this free parameter has significant control over the final accelerated electron spectrum. For the event of Fig. 2, the selected number of transients ($n = 3$) is consistent with the observations (in Fig. 2a, b, MMS observed three transient structures in the foreshock region), and provides the best fit to the observed electron spectrum.

Solar wind electron acceleration from $10 - 100$ eV energies to about 200 keV requires $50 - 100$ scatterings and reflections from the foreshock transients, and each such reflection is a probabilistic process. Therefore, the simulation should contain a sufficient number of test particles to provide good statistics of low-probability multiple reflections, corresponding to the most accelerated electron population. Figure 8b shows that the number of electrons reaching 200 keV is about $\times 10^{-6}$ smaller than the number of core electron population $10 - 100$ eV. In our simulation setup, we consider $5 \times 10^{7}$ test particles to describe well the tail of the electron energy spectrum. Note this tail with about 200 keV energies is mostly formed by the core solar wind distribution, $[10, 100]$ eV. Although the probability of $> 100$ eV to be trapped and further accelerated to higher energies is expected to be higher than for $< 100$ eV electrons, Fig. 8d shows that this hot solar wind population has too low fluxes to contribute to the 100 keV population, i.e., in the solar wind spectrum the number of particles decreases with the energy increase much faster than the probability to be trapped and accelerated in the foreshock increases with the energy.

## Data availability
The observational data from the THEMIS mission shown in the study are publicly available at https://themis.ssl.berkeley.edu/data/themis, and the observational data from the MMS mission shown in the study are publicly available at https://lasp.colorado.edu/mms/sdc/public/about/browse-wrapper/. Source data have been deposited in Zenodo (https://doi.org/10.5281/zenodo.14057804). The datasets generated during and/or analyzed during the current study are available from the corresponding author upon request.

## Code availability
Data analysis was done using Space Physics Environment Data Analysis Software (SPEDAS) V4.1, available at https://spedas.org/. Electron resonant scattering rates models mentioned in the section "Electron resonant scattering rates" are available at https://doi.org/10.5281/zenodo.14252438. Details on the numerical simulations are discussed within the text and are performed using Julia language, publicly available at https://julialang.org/, and the analysis can also be reproduced using a code written in other programming languages with the information given in the section "Numerical simulation approach". The numerical simulation code is available from the corresponding author upon request.

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

## Acknowledgements

We are thankful to the THEMIS and MMS teams and instrument principal investigators for excellent data making possible this study. X.S, A.A., and V.A. acknowledge THEMIS Contract No. NAS5-02099 and NASA Grants 80NSSC22K1634 and 80NSSC21K0581. T.Z.L. acknowledges NSF award AGS-2247760 and NASA grant 80NSSC23K0086. Some of the work was supported by the Geospace Environment Modeling (GEM) Focus Group entitled, "Particle Heating and Thermalization in Collisionless Shocks in the Magnetospheric Multiscale Mission (MMS) Era," led by L.B. Wilson III. We thank the contribution of Emmanuel Masongsong for helping with the first illustration figure.

## Author contributions

X.S. contributed to the conceptualization of the study, data analysis, data interpretation, numerical simulations, and the manuscript's writing. A.A. conducted the theoretical model. V.A., T.Z.L., and L.B.W. contributed to analyzing observations and simulations data, and to data interpretation. All authors participated in the manuscript reviewing and editing.

## Competing interests

The authors declare no competing interests.
