## [Transparent Peer Review file · Nature Communications]

Compound electron acceleration at planetary foreshocks

Corresponding Author: Dr Xiaofei Shi

Version 0:

Reviewer comments:

Reviewer #1

(Remarks to the Author)

The manuscript by Shi et al. submitted to Nature Communications has proposed a model of electron acceleration that may potentially explain energization up to mildly relativistic energies seen around foreshock regions of Earth's bow shock. The model is quite interesting and has not been discussed elsewhere. Qualitatively speaking, the idea itself appears promising to me. However, there are a number of issues that need clarification and more sophistication. Unless these points are reasonably addressed, I would not recommend it for publication in Nature Communications.

Major comments:

- It appears that the authors take a sort of bounce averaging and update the particle pitch angle and energy every half bounce period. Is your L (spatial scale length) a constant over time? If so, is that consistent with the energy gain by Fermi or betatron acceleration? Is the time step Δt energy and pitch-angle dependent as it should be by the definition? These points have to be clarified.
- The energy change due to interaction with whistler waves is not clearly described. I suppose that it should be a diffusion in energy with its coefficient smaller than the pitch-angle diffusion by a factor of $(\omega/kv)^2$, which I presume to be a small correction in the solar wind.
- It is not clear how the authors take into account the particle escape from the system. The description given in the manuscript is rather ambiguous. Please note that the spectral index is theoretically determined by the ratio between the energy gain and escape time scales. In this sense, the escape time scale and its energy dependence should be clearly described.
- The energy gain by SDA is not defined. Similarly, it is not clear how the authors chose $v_f \sim 1000$ km/s as a mean value for Fermi acceleration. In relation to this, the obliquity angle $80^\circ \pm 5^\circ$ mentioned in the paper is hard to believe simply from the appearance of reflected ions and foreshock transients (signatures of a quasi-parallel shock). I was aware that the database by Lalti et al. referred in the paper can be reasonable sometimes but often not. The velocity-magnetic-field mixed coplanarity probably gives a better estimate, but is of course dependent on the chosen time intervals. At least the authors should provide sufficient reasoning for their choice. In any case, since you never know the correct value, it is important to show that the conclusion is not sensitive to the chosen value.
- Observations of energetic electrons indicate that intense fluxes are observed only around high magnetic field strength regions, as is clearly seen in Fig. 2 and also in the earlier paper Wilson et al. (2016). On the other hand, the authors' model is essentially a box model, in which the spatial dependence is averaged out. Does the model appropriately describe the particle acceleration under a certain assumption? If so, what kind of assumption it would be? Also, is the absolute flux something you can directly compare with the observation?
- It is important to recognize that the bow shock has a finite system size. Particles bouncing between the bow shock and foreshock disturbances will be convected by the solar wind (unless the magnetic field is strictly radial), and the particle acceleration time scale should be faster than the convective loss time scale. The authors should comment on this.
- Do you have a gap in 90 degrees in pitch angle for the pitch angle diffusion? If so, does that pose a problem? Or does the reflection by magnetic mirroring resolve it?

Minor comments:

- L.212: The authors state "... field-aligned waves resonating with relativistic electrons ..." I think it would be appropriate state like "... with electrons including the relativistic correction ..." because the authors consider both non-relativistic and relativistic energies.

- Eq.(3) takes average of the pitch-angle diffusion coefficient. In reality, however, this average should take into account particle residence time. In other words, particles stay longer in regions where magnetic field is strong (because of magnetic mirroring). Furthermore, if you have pitch-angle scattering (as certainly you do), the residence time should also be dependent on it because particle transport becomes diffusive. This allows particles to gain more energy than without scattering. I think it may not be necessary to go into such detail at this point, but the authors might consider to take into account such effect in the future.

Reviewer #2

(Remarks to the Author)

In this work, the authors address the interesting problem of electron acceleration in astrophysical systems, relevant to many scientific communities. The focus of the paper is the region upstream of Earth's bow shock, by far the most studied shock from the point of view of direct observations. The authors address the production mechanisms for a population of high-energy, relativistic electrons found in the region where foreshock transients are found upstream of the bow shock. It is shown that the observed electron spectrum in such a region may be reproduced with a model including electron acceleration at the shock as well as at different kinds of foreshock transient and disturbances. The model, main novelty of the work, combines and builds onto existing theories of acceleration at shocks and waves/foreshock transients.

While I find the work very interesting, I believe that it is more suitable for a more specialized journal, as it represents an important but incremental improvement to our understanding of particle acceleration in shock systems. Below, I include a more detailed explanation for such a recommendation.

1. As properly documented by the authors, the finding of such high energy electron population corresponding to foreshock transient, is not new (refs. 8,10,11). Furthermore, the role of foreshock transients in particle acceleration has been documented extensively. Here, different contributions from different structures are merged, in a novel approach that is, in my opinion, not sufficient to grant publication in *Nature Communications*.
2. I find that there is confusion, in the manuscript, between the electron foreshock region, connected to the quasi-perpendicular portion of the Earth's bow shock, and the region studied by the authors. This is an issue, since SDA is efficient for quasi-per shock, upstream of which we are probably less likely to find foreshock transients. This issue is not reported in the submitted work. For example, when addressing the spatial scale of the electron acceleration region (and in Figure S1) the observations do not seem to be conditioned with the geometry observed at the bow shock crossing. One may argue about local departures from the nominal shock normal angle, but these are not documented in the manuscript.
3. While the assumed forms for the diffusion coefficients are well documented by the authors, I do struggle to find novelty in this quasi-linear approach, and the same goes for the Monte-Carlo model (e.g., Ellison and Eichler 1984 and many others), as stated above having the only novelty of including such compound acceleration.
4. If I understood well, no effects of time evolution, crucial for such a small system like Earth's bow shock and transient is included in the model. I think that this is a strong limitation of the study.
5. The link to astrophysical systems and how this study would be relevant to other shocks is, in my opinion, very vague and not argued properly.

Reviewer #3

(Remarks to the Author)

In this Manuscript authors, starting with observations of energetic electron spectra in the foreshock region of the quasi-perpendicular Earth's bow shock, present results coming from a data-constrained model that is able to reproduce observations. The numerical model is based on a probabilistic approach and makes use of the electron scattering rates due to different wave modes (all motivated by observations upstream of the Earth's bow shock). The agreement between the observed power-law electron spectra, with a tail extending up to almost 200 keV, and the energy spectra from the numerical model is emphasized and commented.

The text is very well written and the numerical results in comparison with observations are convincing. The development of a compound acceleration model that reproduces observations represents a step forward in the study of shock acceleration processes. However, I believe there are some points (listed below) that need to be further addressed in order to better understand the multi-step acceleration mechanism the authors invoke. Thus, authors need to revise the text accordingly before the Manuscript can be accepted.

- 1) One aspect that is worth addressing is the influence of the shock geometry (namely the θ_{tabn}) on the acceleration process. From an observational point of view, foreshock transients and relativistic electrons have also been detected at the quasi-parallel Earth's bow shock. For example, Ref. (10) in the Manuscript showed that the presence of large-scale foreshock transients can increase the efficiency of electron acceleration at a quasi-parallel shock. Have authors explored the influence of the shock geometry on the formation of the electron power-law tail in the numerical model?

2) I suggest authors to better specify in the text the values of the characteristic frequencies of the different wave modes considered (for example, in Figure 2-h it is not clear which kind of electrostatic fluctuations the solid thick line in the electric power spectral densities refers to). Discussing the characteristic frequencies of wave modes in relation to the typical Larmor scales/frequencies of energetic electrons would help to understand the scattering mechanism reported in Fig.3.

3) Regarding the analysis of wave modes, can authors explain how they isolate wave modes from a nonlinear turbulent environment? Did they band-pass filter the signal?

4) Lines 187-190: in the estimation of the distance between the upstream transient structures and the Earth's bow shock along the solar wind flow direction, it is not clear to me whether for an event with several transients, as the one reported in Figure 2, the distance from the bow shock is taken for one of them or as an average among the different transients observed. Please, clarify such a point.

5) To what extent does the distance between the bow shock and the transient structure have influence on electron particle acceleration in the model? The authors comment only on the influence of the shock normal speed on the formation of the electron tail in the spectrum (see Figure 4).

Version 1:

Reviewer comments:

Reviewer #1

(Remarks to the Author)

I have read the authors' response and the revised manuscript by Shi et al. that is under consideration in Nature Communications. I found that the authors' revision was not satisfactory, and consequently, I do not recommend it for publication. My opinion is based on the following two reasons: (1) the authors did not address some of the concerns raised in my previous report, and (2) the authors' model may contain physically unreasonable assumptions, which I found difficult to assess from the description given in the paper. The paper would be an interesting contribution to a more specialized journal.

The authors's description of the model is not satisfactory. They did not define the energy gain by SDA ΔE_{SDA} in the Method section (while they did in the response). Also, the relations between $\Delta E_{\text{parallel, Fermi}}$ and ΔE_{Fermi} , $\Delta E_{\text{perp, beta}}$ and ΔE_{Beta} are unclear.

The crucial problem is that they did not describe the problem in sufficient detail. If the model considers bouncing particles between the shock and foreshock transients, Fermi acceleration (if this considers a particle bouncing between the two magnetic walls) should already contain SDA because (classical) SDA is nothing more than a mirror reflection. Similarly, what the authors meant by the betatron acceleration is also unclear. Again, if the particles are bouncing, the magnetic field strength as seen by the particles is essentially unchanged because they are in the (nearly undisturbed) solar wind.

The authors mentioned in the response (but again not in the manuscript) that the accelerated particles may spend more time in regions of compressed magnetic field. This makes the interpretation of the model even more tricky. Remember that SDA is an adiabatic process. During the interaction with the shock, particles will gain energy in the perpendicular direction due to the conservation of the adiabatic moment. In other words, their instantaneous energy gain is due to the betatron. The perpendicular energy is converted to the parallel energy through the reflection. It is hard to understand whether separating SDA and the betatron into two different mechanisms is reasonable.

I might have misunderstood something. The problem in the manuscript is, however, that the description given in the paper is too vague, and there is no way for the readers to judge the validity. Although it might be a reasonable model under some specific conditions, I believe that the information given in the manuscript is far from sufficient.

One final rather minor comment. The authors did not provide a detailed description of the analyses (such as time intervals used) from which they estimated the parameters of the shock. This point is not necessarily critical, but as I mentioned in the previous report, the shock appears to be rather quasi-parallel rather than quasi-perpendicular. Such information should be necessary for reproducibility.

Reviewer #2

(Remarks to the Author)

Thanks for addressing my points. The manuscript has improved and it is certainly interesting, though in my view, the element of novelty justifying publication in Nature Communications is still a bit weak. I have no further comments.

Reviewer #3

(Remarks to the Author)

In the revised version of the Manuscript authors addressed satisfactorily all the major points raised. The methods section now contains additional and detailed information on the data analysis and on the numerical model that give strength to the work.

I believe that the Manuscript can be published in Nature Communications in the present form, since it presents a work that significantly advances our knowledge on electron acceleration at collisionless shocks.

Version 2:

Reviewer comments:

Reviewer #1

(Remarks to the Author)

The authors have significantly improved the quality of the manuscript. Now, at least, the model assumption itself becomes much clearer. I still have a concern about whether the quasi-perpendicular geometry employed by the authors is appropriate or not. However, the idea itself is sufficiently novel, and it would be reasonable to publish the paper for further scrutiny by the community.

Just a minor comment. I believe that both the coplanarity and mass-flux conservation methods require the upstream and downstream intervals. So the authors should state the two time intervals in Methods section.

Response to reviewers' comments on *Compound electron acceleration at planetary foreshocks*

We are grateful to the reviewers's responses. Our point-by-point responses to the comments are listed below in blue and the track changes file is also attached.

Response to the Reviewer # 1

The manuscript by Shi et al. submitted to Nature Communications has proposed a model of electron acceleration that may potentially explain energization up to mildly relativistic energies seen around foreshock regions of Earth's bow shock. The model is quite interesting and has not been discussed elsewhere. Qualitatively speaking, the idea itself appears promising to me. However, there are a number of issues that need clarification and more sophistication. Unless these points are reasonably addressed, I would not recommend it for publication in Nature Communications.

Major Comments:

- *It appears that the authors take a sort of bounce averaging and update the particle pitch angle and energy every half bounce period. Is your L (spatial scale length) a constant over time? If so, is that consistent with the energy gain by Fermi or betatron acceleration? Is the time step dt energy and pitch-angle dependent as it should be by the definition? These points have to be clarified.*

Good point! Indeed, we use the bounce averaging with the time step of energy and pitch-angle updating equal to half of a bounce period. The spatial scale L is shrinking during the simulation to account for the Fermi acceleration, and this effect is taken into account for the evaluation of the bounce period. We have added clarifications to the main text and Methods (in Lines 119-122 and Lines 294-313).

- *The energy change due to interaction with whistler waves is not clearly described. I suppose that it should be a diffusion in energy with its coefficient smaller than the pitch-angle diffusion by a factor of $(\omega/kv)^2$, which I presume to be a small correction in the solar wind.*

Yes, the reviewer is right that the energy diffusion rate is much smaller

than pitch-angle diffusion rate. We have clarified the evaluation of electron energy change due to the resonant scattering by whistler-mode waves in the Method section: using the conservation of electron kinetic energy in the frame moving with the phase velocity of the resonant wave, we calculate the energy change due to resonant interactions from the pitch angle change.

- *It is not clear how the authors take into account the particle escape from the system. The description given in the manuscript is rather ambiguous. Please note that the spectral index is theoretically determined by the ratio between the energy gain and escape time scales. In this sense, the escape time scale and its energy dependence should be clearly described.*

The particles escape the system when their pitch angles are smaller than the loss cone. We also take into account the fact there can be multiple foreshock transient structures in the upstream, and each particle has three chances to be reflected. But if none of the three transients (with amplitudes selected from the probability distribution function shown in Fig. S2(e)) can reflect a specific electron (i.e., its pitch-angle is too small and falls within the loss-cone of all three transients), this electron is considered lost (escapes from the system and is excluded from the simulation distribution). When a particle escapes from the system, we put a new particle at solar wind thermal energy. This procedure has been clarified in the Method section in Lines 264-267.

- *The energy gain by SDA is not defined. Similarly, it is not clear how the authors chose $v_f \sim 1000$ km/s as a mean value for Fermi acceleration. In relation to this, the obliquity angle $80^\circ \pm 5^\circ$ mentioned in the paper is hard to believe simply from the appearance of reflected ions and foreshock transients (signatures of a quasi-parallel shock). I was aware that the database by Lalti et al. referred in the paper can be reasonable sometimes but often not. The velocity-magnetic-field mixed coplanarity probably gives a better estimate, but is of course dependent on the chosen time intervals. At least the authors should provide sufficient reasoning for their choice. In any case, since you never know the correct value, it is important to show that the conclusion is not sensitive to the chosen value.*

To estimate the energy gain from SDA, we use the de-Hoffmann–Teller frame (HTF), where the electric field vanishes ($\vec{E} = -\vec{u} \times \vec{B} = 0$) both upstream and downstream and the particle’s kinetic energy is also conserved. For electrons that can be reflected from the shock due to the mirror force,

their pitch angles should be greater than the loss-cone angle. This reflection results in a finite velocity increase which can be expressed as $\Delta v = 2u_{sh}$, where $u_{sh} = u_0 \cos \theta_{Bn}$ is the upstream plasma flow speed measured in the HTF. This acceleration preferentially increases the parallel energy (the direction along the magnetic field line) of a reflected particle, and therefore, leads to a decrease in the pitch angle. We have added clarification of this energy gain in the Method section.

For Fermi acceleration, the energy gain is also the result of a finite velocity increase during the reflection from the transient boundary moving toward the bow shock. Based on previously published estimates of the transient motion, we use for this boundary the same range of values as for the bow shock speed.

The reviewer is correct that determining the exact bow shock speed is challenging and it may change within the time-scale of simulations. To reduce the effect of this uncertainty on simulation results, we use a random velocity for each electron-shock interaction based on a probability distribution rather than a fixed velocity. The mean value of this distribution does not affect simulation results mostly because it determines the time-scale of acceleration, but does not change the final spectrum (especially when the system reaches steady state of a balance between losses and accelerations). We have added these clarifications to the text.

- *Observations of energetic electrons indicate that intense fluxes are observed only around high magnetic field strength regions, as is clearly seen in Fig. 2 and also in the earlier paper Wilson et al. (2016). On the other hand, the authors' model is essentially a box model, in which the spatial dependence is averaged out. Does the model appropriately describe the particle acceleration under a certain assumption? If so, what kind of assumption it would be? Also, is the absolute flux something you can directly compare with the observation?*

This is a very good point! Indeed, the box model assumes that energetic particles should present everywhere. However, there can be a motion effect: around strong magnetic field gradients most of energetic electrons experience reflection and significantly decrease (reverse) their field-aligned velocities, which means that within this region electrons will spend much more time than in the middle between the bow shock and the foreshock transient. Moreover, large-amplitude magnetic field fluctuations within this region may temporally trap energetic electrons, further increasing their residence time

and fluxes. This speculation cannot be verified within the proposed model and requires more detailed simulations, e.g. massive test particle simulation in the realistic foreshock from the global hybrid model with wave-particle effects included within the stochastic differential approach. In our study we focus on reproducing electron energy spectrum, but leave the spatial distribution of electrons for the next investigation.

Regarding the actual (absolute) fluxes: at the low energy boundary our model should be normalized on the solar wind flux magnitude, and thus with the same energy slope this model, by definition, should repeat observations of the absolute flux values for the entire spectrum.

- *It is important to recognize that the bow shock has a finite system size. Particles bouncing between the bow shock and foreshock disturbances will be convected by the solar wind (unless the magnetic field is strictly radial), and the particle acceleration time scale should be faster than the convective loss time scale. The authors should comment on this.*

The Reviewer is right that the time scale for acceleration should be shorter than that for convective loss. Simulations indicate that electrons can achieve energies of ~ 100 keV around 10 min. The convection speed of foreshock disturbances is considerably slower than the solar wind speed, and is about ~ 100 km/s. Therefore, the convection distance for 10 min is about $10R_E$, twice larger than the averaged distance between foreshock transients and bow shock. However, electron scattering by waves and reflection from the bow shock should result in electron cross-field transport, which allows some portion of electrons to stay longer within the foreshock region. These speculations, of course, must be verified with full test particle simulations in realistic (e.g., derived from the global hybrid simulations) foreshock electromagnetic fields. We have included a note about this model assumption into the text.

- *Do you have a gap in 90 degrees in pitch angle for the pitch angle diffusion? If so, does that pose a problem? Or does the reflection by magnetic mirroring resolve it?*

When electrons reach a very high pitch angle, their parallel velocity is slow and the bouncing period for them will be quite long (even exceeding the simulation time). These electrons are considered to be totally trapped, but their population is not large. Reviewer is right that such high pitch-angle electrons do not experience pitch-angle scattering by waves, but adiabatic reflection changes their pitch-angles.

Minor Comments:

- L.212: The authors state "... field-aligned waves resonating with relativistic electrons ..." I think it would be appropriate state like "... with electrtrons including the relativistic correction ..." because the authors consider both non-relativistic and relativistic energies.

Thanks for pointing out! We have corrected this sentence.

- Eq.(3) takes average of the pitch-angle diffusion coefficient. In reality, however, this average should take into account particle residence time. In other words, particles stay longer in regions where magnetic field is strong (because of magnetic mirroring). Furthremore, if you have pitch-angle scattering (as certainly you do), the residence time should also be dependent on it because particle transport becomes diffusive. This allows particles to gain more energy than without scattering. I think it may not be necessary to go into such detail at this point, but the authors might consider to take into account such effect in the future.

We agree with the Reviewer's observation that diffusion is dependent on the residence time. In our model, we define $dt = \int_L ds/v_{\parallel}$ to represent the time-scale electrons spend near the foreshock transient boundary. This formulation assumes a magnetic mirror configuration and accounts for variations in parallel electron velocity v_{\parallel} . However, as the Reviewer pointed out, we did not thoroughly consider the variation of resident time due to the scattering effects along this trajectory. We will try to address this aspect in future work!

Response to the Reviewer # 2

In this work, the authors address the interesting problem of electron acceleration in astrophysical systems, relevant to many scientific communities. The focus of the paper is the region upstream of Earth's bow shock, by far the most studied shock from the point of view of direct observations. The authors address the production mechanisms for a population of high-energy, relativistic electrons found in the region where foreshock transients are found upstream of the bow shock. It shown that the observed electron spectrum in such a region may be reproduced with a model including electron acceleration at the shock as well as at different kinds of foreshock transient and disturbances. The

model, main novelty of the work, combines and builds onto existing theories of acceleration at shocks and waves/foreshock transients. While I find the work very interesting, I believe that it is more suitable for a more specialized journal, as it represents an important but incremental improvement to our understanding of particle acceleration in shock systems. Below, I include a more detailed explanation for such a recommendation.

- *As properly documented by the authors, the finding of such high energy electrons population corresponding to foreshock transient, is not new (refs. 8,10,11). Furthermore, the role of foreshock transients in particle acceleration has been documented extensively. Here, different contributions from different structures are merged, in a novel approach that is, in my opinion, not sufficient to grant publication in nature Communications.*

Reviewer correctly noted that similar observations of accelerated electron populations have been reported previously. However, in all these publications Authors underlined that there is no existing explanation, especially quantitative explanation, of formation of such populations. There are no more than general speculations about possible role of foreshock transients in electron trapping/acceleration in published literature. In our study we provide the first quantitative (reproducing the spectrum shape of accelerated electrons) model of electron acceleration. This model combines multiple physical effects (adiabatic reflection, wave-particle resonant interactions for different modes, classical shock drift acceleration, and adiabatic heating) and show their relative contribution to the formation of accelerated electron spectra. We believe this result is quite new and deserves consideration for publication.

- *I find that there is confusion, in the manuscript, between the electron foreshock region, connected to the quasi-perpendicular portion of the Earth's bow shock, and the region studied by the authors. This is an issue, since SDA is efficient for quasi-per shock, upstream of which we are probably less likely to find foreshock transients. This issue is not reported in the submitted work. For example, when addressing the spatial scale of the electron acceleration region (and in Figure S1) the observations do not seem to be conditioned with the geometry observed at the bow shock crossing. One may argue about local departures from the nominal shock normal angle, but these are not documented in the manuscript.*

This is a very good point! The shock geometry is the key factor for the formation of the foreshock region, the primary host for electron acceleration

in our model. However, there is uncertainty in determining θ_{bn} (shock normal angle) when using single-spacecraft data due to the spacecraft's trajectory. This means we can encounter a quasi-parallel geometry, then pass through a quasi-perpendicular shock, and exit into a quasi-parallel downstream region. Additionally, collisionless shocks are not stationary and do not act like a simple planar surface or discontinuity. Therefore, determining the exact bow shock angle and speed is challenging and it may change within the time-scale of simulations. To reduce the effect of this uncertainty on simulation results, instead of a single value for shock velocity we use a velocity probability distribution function and determine a specific value of velocity for each electron-shock interaction. We have added a short discussion about this in the Method section.

Although the field-aligned beams of electrons are usually separate from ions, electrons can still be energized and reflected at the quasi-parallel shock. Energetic electron enhancements seen within foreshock transients suggest this. These electrons are within a region that cannot be entirely enclosed by quasi-parallel geometry (i.e., they are often quasi-perpendicular on the sunward side of the outer boundary but will be quasi-parallel in other parts).

- *While the assumed forms for the diffusion coefficients are well documented by the authors, I do struggle to find novelty in this quasi-linear approach, and the same goes for the Monte-Carlo model (e.g., Ellison and Eichler 1984 and many others), as stated above having the only novelty of including such compound acceleration.*

To our best knowledge the quasi-linear diffusion model has not been applied before to the system with so strong background magnetic field and plasma fluctuations. In our model we have incorporated the effect of resonant broadening for quasi-linear diffusion due to these fluctuations: the fluctuation-averaged diffusion rates cover much wider energy and pitch-angle ranges than such rates evaluated for static (averaged) background conditions. This fluctuation-averaging allows us to describe effective scattering of electrons in a wide energy range, the key result for electron trapping and continuous acceleration to 200keV. Together with the inclusion of multiple wave modes (e.g., electrostatic turbulence constituting of nonlinear electron-scale structures has never been included into simulations of electron dynamics near the shock waves before), this fluctuation-averaging is a quite unique theoretical solution proposed and incorporated in our study.

- *If I understood well, no effects of time evolution, crucial for such a small system like Earth's bow shock and transient is included in the model. I think that this is a strong limitation of the study.*

Our model includes the time evolution effects associated with the foreshock transient motion that (1) changes time-scales of electron bouncing (which affects the resonant electron scattering), (2) drives Fermi acceleration. We have included clarifications of this dynamical effects into the Method section.

- *The link to astrophysical systems and how this study would be relevant to other shocks is, in my opinion, very vague and not argued properly.*

Reviewer is right, that we provided only a brief speculation of possible application of our model to astrophysical systems. This is mostly because our model requires specification of multiple wave characteristics unavailable for such systems. We have included a note that possible generalization of our model to astrophysical systems must be supported by numerical simulations of wave activity (wave modes) operating in such systems.

Response to the Reviewer # 3

In this Manuscript authors, starting with observations of energetic electron spectra in the foreshock region of the quasi-perpendicular Earth's bow shock, present results coming from a data-constraint model that is able to reproduce observations. The numerical model is based on a probabilistic approach and makes use of the electron scattering rates due to different wave modes (all motivated by observations upstream of the Earth's bow shock). The agreement between the observed power-law electron spectra, with a tail extending up to almost 200 keV, and the energy spectra from the numerical model is emphasized and commented. The text is very well written and the numerical results in comparison with observations are convincing. The development of a compound acceleration model that reproduces observations represents a step forward in the study of shock acceleration processes. However, I believe there are some points (listed below) that need to be further addressed in order to better understand the multi-step acceleration mechanism the authors invoke. Thus, authors need to revise the text accordingly before the Manuscript can be accepted.

- *One aspect that is worth addressing is the influence of the shock geometry (namely the θ_{bn}) on the acceleration process. From an observational point of view, foreshock transients and relativistic electrons have also been detected at the quasi-parallel Earth's bow shock. For example, Ref. (10) in the Manuscript showed that the presence of large-scale foreshock transients can increase the efficiency of electron acceleration at a quasi-parallel shock. Have authors explored the influence of the shock geometry on the formation of the electron power-law tail in the numerical model?*

This is very good point! Indeed, the shock geometry is the key factor for the formation of the foreshock region, the primary host for electron acceleration in our model. Moreover, θ_{bn} determines the efficiency of SDA mechanism contributing to electron acceleration. Our model assumes the existence of the foreshock region, i.e. existence of reflected ion beams interacting with convected discontinuities, and this assumption cannot be changed within the model concept. However, we have added a short discussion about the importance of the shock configuration for the proposed acceleration mechanism (see Methods).

- *I suggest authors to better specify in the text the values of the characteristic frequencies of the different wave modes considered (for example, in Figure 2-h it is not clear which kind of electrostatic fluctuations the solid thick line in the electric power spectral densities refers to). Discussing the characteristic frequencies of wave modes in relation to the typical Larmor scales/frequencies of energetic electrons would help to understand the scattering mechanism reported in Fig.3.*

Thanks for the suggestions! We have added the typical frequencies and more wave characteristics in the caption of Figure 2 and also added the discussion about frequency, wavelengths, and electron resonance in the main text in Lines 98-118.

- *Regarding the analysis of wave modes, can authors explain how they isolate wave modes from a nonlinear turbulent environment? Did they band-pass filter the signal?*

Yes, for whistler-mode wave modes, we applied a band-pass filter to extract the wave signals. For high-frequency whistler waves, a 0.5-second window (encompassing approximately 10 wave periods) was used for the band-pass filtering to capture the wave properties. For low-frequency magnetosonic

waves, we initially computed the averaged background magnetic field over a 30-second interval to determine the averaged frequency range for the band-pass filter, and then extracted the wave signal. For electrostatic turbulence we use electric field (potentials) measured by different spacecraft antennas to isolate spatially localized electric field structures and analyse their properties.

- *Lines 187-190: in the estimation of the distance between the upstream transient structures and the Earth's bow shock along the solar wind flow direction, it is not clear to me whether for an event with several transients, as the one reported in Figure 2, the distance from the bow shock is taken for one of them or as an average among the different transients observed. Please, clarify such a point.*

We first calculate the distance for all events where two satellites are involved: one satellite observes a transient in the upstream region while the other simultaneously crosses the bow shock. This gives us a statistic of the distances, dx . From this statistic, we obtain a distribution of distances for each individual transient structure, as shown in Figure S1(g). In the simulation, to generalize the results, we use the average distance across all transient structures. We have included a detailed description of this procedure into the Method section.

- *To what extent does the distance between the bow shock and the transient structure have influence on electron particle acceleration in the model? The authors comment only on the influence of the shock normal speed on the formation of the electron tail in the spectrum (see Figure 4).*

This is a good question. The distance between the bow sock and the transient may influence the acceleration process if the transient convection time is longer than the simulation time, i.e. if electrons can spend all the time of simulation between the bow shock and the same transient. However, in reality, electrons frequently escape from one transient's magnetic trapping and get caught between the bow shock and another transient. These transitions occur often in our simulations, making the individual transient characteristics (like the distance to the bow shock) less significant due to the random nature of electron interactions with transients. To confirm this, we have included Figure S3 (c) which shows that the simulation results remain consistent even when we vary the mean distance, dx , between the bow shock and the transient.

Track Changes of *Compound electron acceleration at*
*planetary foreshocks*

Xiaofei Shi^{1*}, Anton Artemyev¹, Vassilis Angelopoulos¹, Terry Liu¹, and Lynn B.
Wilson III²

¹Department of Earth, Planetary, and Space Sciences, University of California, Los
Angeles, California, 90095, USA

²NASA Goddard Space Flight Center, Heliophysics Science Division, Greenbelt,
MD20771, USA

* Corresponding Author: Xiaofei Shi, sxf1698@g.ucla.edu

July 24, 2024

[revised manuscript text omitted]

Trapping can be accomplished by adiabatic reflection or electron pitch-angle scattering as follows:
 First, the foreshock region is filled with transients with large magnetic field fluctuations^{23,25}. The
 compressional nature of these fluctuations may allow them to adiabatically reflect electrons, trapping
 them between the bow shock and the ensemble of transients within the foreshock (**such electrons**
 **will be bouncing**). Second, the foreshock magnetic boundaries host intense electrostatic turbu-
 lence^{17,18,16} (Fig. 2 (h)), high-frequency whistler mode waves^{19,20} (Fig. 2 (f)) , and low-frequency
 magnetosonic waves^{21,22} (Fig. 2 (g)). These wave modes provide effective pitch-angle scattering for
 electrons covering a wide energy range. Figure 3(a-c) shows typical waveforms of these wave-modes.
 **Electrostatic turbulence is dominated by ion acoustic solitary waves, ion holes, and electron holes,**
 **all having spatial scales about Debye length and propagation velocities within the range between ion**
 **to electron thermal speeds, which are $[10^{-3}, 10^{-4}]$ times of the speed of light^{17,18,16}. High-frequency**
 **electromagnetic whistler mode waves have spatial scales (wavelength) about the electron inertial**
 **length and propagation speed of $\sim 10^{-2}$ of the speed of light^{19,20}. Low-frequency magnetosonic**
 **waves have spatial scales (wavelength) about the ion inertial length and propagate at a speed of**
 **$[10^{-3}, 10^{-4}]$ of the speed of light²². The efficiency (rate) of electron pitch-angle scattering by these**
 **waves is given by the bounce-averaged pitch-angle diffusion rate ($D_{\alpha\alpha}$, in rads^2/s , also see **Meth-****
 **ods**). The mechanics for calculating the diffusion rates for these waves are well-established^{27,28,18}
 for a homogeneous or weakly inhomogeneous magnetic field. However, the foreshock region is also
 filled with large amplitude magnetic field fluctuations which provide strong inhomogeneity **along**
 **electron bounce trajectories**.. Thus, we averaged the standard rates of pitch-angle diffusion over
 an ensemble of observed magnetic field fluctuations, using THEMIS statistics (see **Methods**) to
 establish the spatial scales of these fluctuations. **Figures 3(d-f) show such averaged rates: electro-**
 **static turbulence (EST) with frequencies above the electron cyclotron frequency (f_{ce}) mostly scatters**
 **$< 1\text{keV}$ electrons of large and intermediate pitch-angles, high-frequency whistler mode waves (WW)**
 **with frequencies between low-hybrid frequency (f_{lh}) and f_{ce} mostly scatter $< 10\text{keV}$ electrons with**
 **the resonant energy increasing with pitch-angle, and magnetosonic waves (MSW) with frequency**
 **between ion cyclotron frequency and f_{lh} mostly scatter $> 10\text{keV}$ field-aligned electrons. Figure 3**
 **confirms that the three wave modes cover five orders of magnitude in energy, from the solar wind**
 **electron energy $\sim 10\text{eV}$ to near the maximum observed energy of energetic electrons $\sim 100\text{keV}$.**

The acceleration mechanism includes electron SDA at the bow shock (**determined by the shock**
 **speed; see **Methods****), and Fermi and betatron electron acceleration upstream of it (**determined**
 **by shrinking of the spatial scale of electron bouncing motion and by magnetic field increase within**
 **the core region; see **Methods****), due to the foreshock transient motion²⁹. Note that although high-
 frequency whistler mode waves also contribute to electron acceleration in our model²⁰, the main
 role of all three wave modes and compressional fluctuations is to trap electrons near the bow shock,
 allowing them to experience multiple SDA and adiabatic (Fermi and betatron) acceleration.

To reproduce the observed electron spectrum, we simulate electron dynamics affected by a combi-
 nation of electron scattering and acceleration (the numerical methods are expounded in **Methods**).
 We start the simulation with the solar wind electron distribution (grey color) where most of the
 electrons have energies below 100eV (although there is a finite population up to 1keV), and aim
 to resolve the question of electron acceleration up to 100 – 200keV (see Fig. 4). When we include
 only SDA and adiabatic reflection from the compressional magnetic field fluctuations of foreshock
 transients, most electrons escape the foreshock region after ~ 7 reflections from the shock, and

Fig. 2: Observations of flux enhancement of tens to hundreds of keV electrons at a foreshock transient. Between 00:30 and 02:30 UT, MMS crossed Earth’s bow shock and foreshock, where multiple foreshock transients were detected. Panels (a) and (b) show magnetic field strength and ion energy spectra, respectively. Panels (c-h) zoom into a subset of the above interval during a foreshock transient event. From top to bottom these panels show: (c) the magnetic field strength, (d) the ion energy spectrum, (e) the electron energy spectrum for 50 – 200 keV electrons indicating the presence of relativistic electrons up to 150keV, (f, g) magnetic field power spectra for high and low frequencies, respectively; the white lines in (f,g) are electron cyclotron frequency and low hybrid frequency, respectively, (h) the high-frequency electric field power spectrum; electrostatic turbulence observed around the electron cyclotron frequency (black lines), but mainly below ion plasma frequency (white line). Panel (i) illustrates the observed electron phase space density (df) at the upstream region outside transients (magenta lines) and at the foreshock transient of panels (c-h) (blue lines). Reliable measurements are limited to data above the 1 count level (dashed red line). Notably, the electron df during the enhancement adheres to a power-law behavior, with df proportional to E^{-4} (dashed blue line).

the maximum acceleration does not exceed 1 – 3keV (magenta curve). If EST scattering is added,
 electrons may experience up to ~ 12 reflections from the shock and gain 3 – 5keV (red curve). The
 inclusion of electron WW scattering and acceleration increases the number of electron reflections
 from the shock further, up to ~ 24 , with the electron energy gain reaching 5 – 7keV (green curve).
 Scattering of more than a few keV electrons requires the inclusion of MSW scattering, which in-
 creases the number of electron reflections from the bow shock to ~ 40 , whereas the electron energy
 gain reaches 10 – 20keV (blue curve). Therefore, a combination of SDA, adiabatic reflection from
 the foreshock transients, and resonant scattering by three different wave modes can provide acceler-
 ation of $< 100\text{eV}$ solar wind electrons to $\sim 20\text{keV}$. The next effect to be included is the Fermi and
 betatron adiabatic acceleration. The leading edge of the foreshock transient often forms its shock
 wave propagating upstream ahead of the bow shock with a velocity comparable to that of the bow
 shock. Electrons trapped between two shocks moving toward each other (bow shock and transient
 shock) experience Fermi acceleration³⁰. As the foreshock transient moves (collapses) onto the bow
 shock, its magnetic field is compressed, increasing by a factor of ~ 3 (see¹⁰). This effect should
 provide electron betatron heating¹⁰. Figure 4 shows that when both Fermi and betatron adiabatic
 (AD) acceleration are also included in the simulation, the electron spectrum reaches $\sim 200\text{keV}$ and
 attains a power law falloff $\sim E^{-4}$ (black curve). The variability of the model output spectrum due
 to uncertainties in the bow shock speed determination is depicted by the blue-shaded region (see
 **Methods** for details).

The good agreement of the model results of Fig. 4 with observations in Fig. 2(i) validates the
 proposed scenario of solar wind electron acceleration in the foreshock to 100 – 200keV. Such accel-
 eration transforms a small subset of the initial $< 1\text{keV}$ solar wind electron distribution (comprising
 a core population below 100eV and an exponential energy tail) into a power law distribution with
 an E^{-4} falloff, a power law in the 1-200keV energy range. Importantly, efficient electron accel-
 eration from solar wind energies, $\sim 10 - 100\text{eV}$, up to near-relativistic energies, $\sim 200\text{keV}$, cannot be
 otherwise explained by a single mechanism of electron scattering. Rather, multiple wave modes, adi-
 abatic reflections, and adiabatic Fermi and betatron acceleration compound the energy gain arising

[revised manuscript text omitted]

 bilistic approach^{44,45}, which is similar to the Monte-Carlo approach usually applied to astrophysical
 shock waves^{46,47}. An elementary model time-step of electron dynamics is the bounce period between
 the bow shock and the foreshock transient, $\tau_b \approx \ell/v_{\parallel}$ where ℓ is a spatial scale of bouncing. If the
 electron pitch-angle is sufficiently small to cross the bow shock or the foreshock transient boundary,
 it is considered to be lost and a new electron with the solar wind energy substitutes this lost electron
 in the simulation. The bow shock magnetic field has a spatial gradient increasing it ~ 4 times from
 the core field and we do not change the shock configuration for the entire simulation, whereas for
 each electron interaction with the foreshock transient boundary we generate the transient boundary
 magnetic field $B_{boundary}/B_{core}$ from the distribution function of $B_{boundary}/B_{core}$ obtained in MMS
 statistics (see Fig. **S2**(e)). We assume there can be three transient structures during a time interval
 in the foreshock⁴⁰, so electrons have three chances to be reflected by the transient boundary (see
 discussion below). Although the number of particles is conserved within the simulation, we also
 assume that the core of solar wind electron distribution remains unchanged (as there is an almost
 infinite source of for this core), and in Figure 4 we add a small population of solar wind electrons
 with < 100 eV energy to all spectra to make them the same for < 100 eV energy range.

Within one bounce period each electron has the opportunity to experience the combined effects
 of shock drift acceleration (SDA), pitch-angle scattering induced by electrostatic turbulence, high-
 frequency whistler waves, and magnetosonic waves. Additionally, electron energy can be changed by
 wave-particle resonant interactions with high-frequency whistler-mode waves, Fermi acceleration due
 to reflection from the moving boundary of the foreshock transient, and betatron acceleration resulting
 from the increase in the core magnetic field due to plasma compression by the moving foreshock
 transient^{10,29}. The equations describing electron energy E and pitch-angle α recalculation within

one bounce period (between n and $n + 1$ bounces) are:

$$\begin{aligned}
\alpha_{n+1/2} &= \alpha_n + \sum_{i=EST,WW,MSW} W_i \sqrt{\langle D_{\alpha\alpha}(E_n, \alpha_n) \rangle_i} \\
E_{\parallel, n+1/2} &= E_{\parallel, n} + \Delta E_{\parallel, WW}(E_n, \alpha_n + 1/2) \\
E_{\perp, n+1/2} &= E_{\perp, n} + \Delta E_{\perp, WW}(E_n, \alpha_n + 1/2) \\
E_{\parallel, n+1} &= E_{\parallel, n+1/2} + \Delta E_{\text{Fermi}}(E_{\parallel, n+1/2}) + \Delta E_{\text{SDA}}(E_{\parallel, n+1/2}) \\
E_{\perp, n+1} &= E_{\perp, n+1/2} + \Delta E_{\text{Beta}}(E_{\perp, n+1/2}) \\
\alpha_{n+1} &= \arctan(E_{\perp, n+1}/E_{\parallel, n+1})
\end{aligned} \tag{4}$$

where $W_i = \mathcal{N} \cdot dt$, and $\mathcal{N}(0, 1)$ is a random number from the normal Gaussian probability dis-
tribution with a zero mean value and unity dispersion⁴⁴, $dt = \int_L ds/v_{\parallel}$ represents the time-scale
electrons spend around the foreshock transient boundary (where all wave modes are hosted), with L
denoting the spatial scale of this boundary. The electron energy conservation in the reference frame
of the resonant wave provides a direct relation between pitch-angle and energy diffusion rates²⁸. We
use this relation to calculate the electron energy change caused by resonant interactions between
high-frequency whistler-mode waves and electrons (ΔE_{WW}) through pitch-angle change (WW part
of $\alpha_{n+1/2} - \alpha_n$). The approach to evaluate the pitch-angle and energy changes using Eq. (4) assumes
bounce averaging due to the cumulative effect of changes over one bounce period.

Fermi acceleration (due to shrinking of ℓ scale) and SDA change the electron parallel velocity
(energy) component, given by $\Delta E_{\parallel, \text{Fermi}} = 2m_e v_f^2$, where v_f is the bow shock and foreshock transient
boundary speed⁴⁸. Determination of this velocity for specific shock can be associated with large
uncertainties, and this velocity may change within the simulation time. Therefore, to reduce the
effect of such uncertainties on our results, instead of specific v_f value we determine v_f for each
interaction from the normal Gaussian probability distribution with a mean value of 1000 km/s and
a standard deviation of 400 km/s, which correspond to the shock speed ~ 100 km/s (see⁴⁹) and shock
normal angle above 85° (see⁵⁰). We have verified that the change of the mean value to 800km/s and
1200km/s does not change the final spectrum of accelerated electrons. Betatron acceleration changes
the perpendicular component as $\Delta E_{\perp, \text{beta}} = E_{\perp, n} ((B + \delta B)/B - 1)$, where δB is the core magnetic
field increment for dt time-scale (we consider $\sum \delta B/B = 3$ for the entire simulation period). For each
electron, the simulation starts with the bouncing spatial scale ℓ selected from the dx distribution
shown in Fig. S1(g). Over time, ℓ shrinks (lead to Fermi acceleration), affecting the electron bounce
time τ_b . If an electron escapes from trapping with a transient structure or when such a structure
reaches the bow shock, the electron may start bouncing between the shock and another structure. In
this case, ℓ is selected again from the dx distribution. At the end of each bounce period, we check if
the electron can continue to the next period. This is determined by whether the electron pitch-angle
is large enough to be outside the loss-cone of the bow shock and three foreshock transients. If the
pitch-angle is too small, the electron is excluded from the system and replaced by a new electron
with solar wind energy.

There are two key assumptions regarding the bow shock and foreshock configuration that we
need to discuss. First, our simulation spans about ~ 10 minutes of real time, which aligns closely
with (or slightly larger than) the convection time required for the foreshock transient to traverse the
distance dx shown in Fig. S1(g) at the typical convection speed⁹. To maintain the trapped electrons
within the foreshock for these 10 minutes, additional cross-field electron transport, likely caused by
wave scattering and the bow shock, would be necessary, though quantifying this transport is beyond
our simulation's scope. These electrons can then be trapped by another transient, with up to three
such trappings possible in a single bounce period. Thus, the properties of individual transients do

not significantly affect the acceleration. Figure **S3** (c) shows that the change of the mean distance
dx does not change the final spectrum of acceleration electrons. Second, our acceleration model
presumes the existence of the foreshock, which necessitates reflected ion beams and a particular
shock normal angle. This assumption prevents us from exploring how acceleration efficiency varies
with the shock normal angle. Therefore, further studies using more advanced simulations are needed
to quantify the roles of cross-field transport and shock configuration in electron acceleration.

Most of the system parameters for numerical simulations of electron dynamics, scattering, and
acceleration are selected according to spacecraft statistical observations. However, the role of a free
model parameter, the number of simultaneously existing foreshock transients, requires additional
verification. The number of such transients determines the probability of electron adiabatic reflection
in the foreshock and thus should affect the electron acceleration efficiency. Figure **S3**(a) shows that
with other system parameters fixed, the increase/decrease of the number of foreshock transients
($n = 3 \pm 1$) results in a variation of phase space density of $> 10\text{keV}$ electrons and maximum energies
within $\in [75, 300]\text{keV}$. Therefore, this free parameter has significant control over the final accelerated
electron spectrum. For the event of Fig. 2 the selected number of transients ($n = 3$) is consistent
with the observations (in Fig. 2 (a,b), MMS observed three transient structures in the foreshock
region), and provides the best fit to the observed electron spectrum.

Solar wind electron acceleration from $10 - 100\text{eV}$ energies to $\sim 200\text{keV}$ requires $\sim 50 - 100$ of
scatterings and reflections from the foreshock transients, and each such reflection is a probabilistic
process. Therefore, the simulation should contain a sufficient number of test particles to provide
good statistics of low-probability multiple reflections, corresponding to the most accelerated electron
population. Figure **S3**(b) shows that number of electrons reaching $\sim 200\text{keV}$ is about $\times 10^{-6}$ smaller
than the number of core electron population $\sim 10 - 100\text{eV}$. In our simulation setup, we consider
5×10^7 test particles to describe well the *tail* of the electron energy spectrum. Note this *tail* with
$\sim 200\text{keV}$ energies is mostly formed by the core solar wind distribution, $[10, 100] \text{eV}$. Although the
probability of $> 100\text{eV}$ to be trapped and further accelerated to higher energies is expected to be
higher than for $< 100\text{eV}$ electrons, Figure **S3**(d) shows that this hot solar wind population has too
low fluxes to contribute to the 100keV population, i.e. in the solar wind spectrum the number of
particles decreases with the energy increase much faster than the probability to be trapped and
accelerated in the foreshock increases with the energy.

Data Availability

All THEMIS and MMS data are publicly available through the Space Physics Data Facility (SPDF)
at <https://cdaweb.gsfc.nasa.gov/>, and via THEMIS and MMS datasets: <http://themis.ssl.berkeley.edu>
and <https://lasp.colorado.edu/mms/sdc/public/about/browse-wrapper/>.

Code Availability

Data analysis was done using Space Physics Environment Data Analysis Software (SPEDAS) V4.1,
available at <https://spedas.org/>. The computer code of numerical simulations in this study is avail-
able upon request to the corresponding author.

Fig. S1: Typical multi-satellite observation of a foreshock transient's plasma environment detected by THEMIS. (a-b) THEMIS-E crossing of the bow shock, denoted by the vertical black line. (c-e) concurrent THEMIS-D (nearby THEMIS-E) observation of the foreshock transient, identified by its magnetic field enhancement and electron density perturbation in the upstream region; (f) locations of THEMIS-D and THEMIS-E in the GSE coordinate system, with the dashed black line indicating the position of the bow shock. We have found approximately 100 similar events, involving one THEMIS satellite crossing the bow shock and another being in the upstream region observing foreshock transient perturbations; (g) statistical distribution of the distances between the bow shock and transient structure, offering insights into the spatial scales of the foreshock acceleration region.

Fig. S2: Derivation of simulation parameters for waves. (a) observed magnetic field power spectrum of magnetosonic waves and high-frequency whistler-mode waves (black lines) and a fit to the observations (blue line); (b) combined diffusion coefficient for the three types of waves calculated at the minimum magnetic field (inside the core); (c) distribution of magnetic field perturbations within foreshock transient structures; (d) combined diffusion coefficient averaged over the background magnetic field perturbations; (e) distribution of the ratio $B_{boundary}/B_{core}$.

Fig. S3: Investigation of the role of multiple foreshock transient crossings in the electron energization. (a) effect of the number of foreshock transients (n_t) on electron spectrum: as n_t increases, electrons can be accelerated to higher energies; (b) normalized number density distribution of the final result of a simulation with 5×10^7 electrons; (c) effect of different distance dx : for a given time (here is 10 min), a larger distance results in fewer bounces because of a longer bouncing period; (d) probability distribution in the (initial energy, final energy) space, demonstrating that the main source of the accelerated particles is the core of the solar wind ($< 100 \text{ eV}$)

References

- [1] Frank C. Jones and Donald C. Ellison. The plasma physics of shock acceleration. *Space Sci.*
*Rev.*, 58(1):259–346, December 1991. doi: 10.1007/BF01206003.
- [2] Joe Giacalone. Particle Acceleration at Shocks Moving through an Irregular Magnetic Field.
*Astrophys. J.*, 624(2):765–772, May 2005. doi: 10.1086/429265.
- [3] Silvia Perri, Andrei Bykov, Hans Fahr, Horst Fichtner, and Joe Giacalone. Recent Developments
in Particle Acceleration at Shocks: Theory and Observations. *Space Sci. Rev.*, 218(4):26, June
2022. doi: 10.1007/s11214-022-00892-5.
- [4] K. Koyama, R. Petre, E. V. Gotthelf, U. Hwang, M. Matsuura, M. Ozaki, and S. S. Holt.
Evidence for shock acceleration of high-energy electrons in the supernova remnant SN1006.
*Nature*, 378(6554):255–258, November 1995. doi: 10.1038/378255a0.
- [5] F. A. Aharonian, A. G. Akhperjanian, K. M. Aye, A. R. Bazer-Bachi, M. Beilicke, W. Ben-
bow, D. Berge, P. Berghaus, K. Bernlöhr, O. Bolz, C. Boisson, C. Borgmeier, F. Breitling,
374 A. M. Brown, J. Bussons Gordo, P. M. Chadwick, V. R. Chitnis, L. M. Chounet, R. Cornils,
375 L. Costamante, B. Degrange, A. Djannati-Ataï, L. O’C. Drury, T. Ergin, P. Espigat, F. Fein-
376 stein, P. Fleury, G. Fontaine, S. Funk, Y. A. Gallant, B. Giebels, S. Gillessen, P. Goret, J. Guy,
C. Hadjichristidis, M. Hauser, G. Heinzelmann, G. Henri, G. Hermann, J. A. Hinton, W. Hof-
mann, M. Holleran, D. Horns, O. C. de Jager, I. Jung, B. Khélifi, Nu. Komin, A. Konopelko,
I. J. Latham, R. Le Gallou, M. Lemoine, A. Lemièrre, N. Leroy, T. Lohse, A. Marcowith,
C. Masterson, T. J. L. McComb, M. de Naurois, S. J. Nolan, A. Noutsos, K. J. Orford, J. L.
Osborne, M. Ouchrif, M. Panter, G. Pelletier, S. Pita, M. Pohl, G. Pühlhofer, M. Punch, B. C.
Raubenheimer, M. Raue, J. Raux, S. M. Rayner, I. Redondo, A. Reimer, O. Reimer, J. Ripken,
383 M. Rivoal, L. Rob, L. Rolland, G. Rowell, V. Sahakian, L. Saugé, S. Schlenker, R. Schlickeiser,
C. Schuster, U. Schwanke, M. Siewert, H. Sol, R. Steenkamp, C. Stegmann, J. P. Tavernet,
C. G. Théoret, M. Tluczykont, D. J. van der Walt, G. Vasileiadis, P. Vincent, B. Visser, H. J.
Völk, and S. J. Wagner. High-energy particle acceleration in the shell of a supernova remnant.
*Nature*, 432(7013):75–77, November 2004. doi: 10.1038/nature02960.
- [6] A. Masters, L. Stawarz, M. Fujimoto, S. J. Schwartz, N. Sergis, M. F. Thomsen, A. Retinò,
H. Hasegawa, B. Zieger, G. R. Lewis, A. J. Coates, P. Canu, and M. K. Dougherty. Electron
acceleration to relativistic energies at a strong quasi-parallel shock wave. *Nature Physics*, 9:
164–167, March 2013. doi: 10.1038/nphys2541.
- [7] N. Dresing, S. Theesen, A. Klassen, and B. Heber. Efficiency of particle acceleration at inter-
planetary shocks: Statistical study of STEREO observations. *Astronomy & Astrophysics*, 588:
A17, April 2016. doi: 10.1051/0004-6361/201527853.
- [8] L. B. Wilson, D. G. Sibeck, D. L. Turner, A. Osmane, D. Caprioli, and V. Angelopoulos.
Relativistic Electrons Produced by Foreshock Disturbances Observed Upstream of Earth’s
Bow Shock. *Physical Review Letters*, 117(21):215101, November 2016. doi: 10.1103/Phys-
RevLett.117.215101.
- [9] D. L. Turner, N. Omid, D. G. Sibeck, and V. Angelopoulos. First observations of foreshock
bubbles upstream of Earth’s bow shock: Characteristics and comparisons to HFAs. *Journal of*
*Geophysical Research (Space Physics)*, 118(4):1552–1570, April 2013. doi: 10.1002/jgra.50198.

- [10] Terry Z. Liu, Vassilis Angelopoulos, and San Lu. Relativistic electrons generated at Earth's
quasi-parallel bow shock. *Science Advances*, 5(7):eaaw1368, July 2019. doi: 10.1126/sci-
adv.aaw1368.
- [11] Takanobu Amano, Yosuke Matsumoto, Artem Bohdan, Oleh Kobzar, Shuichi Matsukiyo, Mit-
suo Oka, Jacek Niemiec, Martin Pohl, and Masahiro Hoshino. Nonthermal electron acceleration
at collisionless quasi-perpendicular shocks. *Reviews of Modern Plasma Physics*, 6(1):29, De-
cember 2022. doi: 10.1007/s41614-022-00093-1.
- [12] Takanobu Amano and Masahiro Hoshino. Theory of Electron Injection at Oblique Shock of
Finite Thickness. *Astrophys. J.*, 927(1):132, March 2022. doi: 10.3847/1538-4357/ac4f49.
- [13] G. M. Webb, W. I. Axford, and T. Terasawa. On the drift mechanism for energetic charged
particles at shocks. *Astrophys. J.*, 270:537–553, July 1983. doi: 10.1086/161146.
- [14] R. A. Treumann. Fundamentals of collisionless shocks for astrophysical application, 1. Non-
relativistic shocks. *The Astronomy and Astrophysics Review*, 17:409–535, December 2009. doi:
10.1007/s00159-009-0024-2.
- [15] Takuma Katou and Takanobu Amano. Theory of Stochastic Shock Drift Acceleration for Elec-
trons in the Shock Transition Region. *Astrophys. J.*, 874(2):119, April 2019. doi: 10.3847/1538-
4357/ab0d8a.
- [16] I. Y. Vasko, F. S. Mozer, S. D. Bale, and A. V. Artemyev. Ion-Acoustic Waves in a Quasi-
Perpendicular Earth's Bow Shock. *Geophys. Res. Lett.*, 49(11):e98640, June 2022. doi:
10.1029/2022GL098640.
- [17] M. Balikhin, S. Walker, R. Treumann, H. Alleyne, V. Krasnoselskikh, M. Gedalin, M. Andre,
423 M. Dunlop, and A. Fazakerley. Ion sound wave packets at the quasiperpendicular shock front.
*Geophys. Res. Lett.*, 32(24):L24106, December 2005. doi: 10.1029/2005GL024660.
- [18] S. R. Kamaletdinov, I. Y. Vasko, R. Wang, A. V. Artemyev, E. V. Yushkov, and F. S. Mozer.
Slow electron holes in the Earth's bow shock. *Physics of Plasmas*, 29(9):092303, September
2022. doi: 10.1063/5.0102289.
- [19] M. Oka, L. B. Wilson, III, T. D. Phan, A. J. Hull, T. Amano, M. Hoshino, M. R. Argall,
O. Le Contel, O. Agapitov, D. J. Gershman, Y. V. Khotyaintsev, J. L. Burch, R. B. Torbert,
C. Pollock, J. C. Dorelli, B. L. Giles, T. E. Moore, Y. Saito, L. A. Avanov, W. Paterson, R. E.
Ergun, R. J. Strangeway, C. T. Russell, and P. A. Lindqvist. Electron Scattering by High-
frequency Whistler Waves at Earth's Bow Shock. *Astrophys. J. Lett.*, 842:L11, June 2017. doi:
10.3847/2041-8213/aa7759.
- [20] Xiaofei Shi, Anton Artemyev, Vassilis Angelopoulos, Terry Liu, and Xiao-Jia Zhang. Evidence
of Electron Acceleration via Nonlinear Resonant Interactions with Whistler-mode Waves at
Foreshock Transients. *Astrophys. J.*, 952(1):38, July 2023. doi: 10.3847/1538-4357/acd9ab.
- [21] D. Krauss-Varban, N. Omidi, and K. B. Quest. Mode properties of low-frequency waves:
Kinetic theory versus Hall-MHD. *J. Geophys. Res.*, 99(A4):5987–6010, April 1994. doi:
10.1029/93JA03202.
- [22] L. B. Wilson, III, A. Koval, A. Szabo, A. Breneman, C. A. Cattell, K. Goetz, P. J. Kellogg,
441 K. Kersten, J. C. Kasper, B. A. Maruca, and M. Pulupa. Observations of electromagnetic

- whistler precursors at supercritical interplanetary shocks. *Geophys. Res. Lett.*, 39:L08109,
April 2012. doi: 10.1029/2012GL051581.
- [23] Y. Hobara, S. N. Walker, M. Balikhin, O. A. Pokhotelov, M. Dunlop, H. Nilsson, and H. Rème.
Characteristics of terrestrial foreshock ULF waves: Cluster observations. *Journal of Geophysical*
*Research (Space Physics)*, 112(A7):A07202, July 2007. doi: 10.1029/2006JA012142.
- [24] L. B. Wilson. Low Frequency Waves at and Upstream of Collisionless Shocks. *Washington DC*
*American Geophysical Union Geophysical Monograph Series*, 216:269–291, February 2016. doi:
10.1002/9781119055006.ch16.
- [25] Hui Zhang, Qiugang Zong, Hyunju Connor, Peter Delamere, Gábor Facskó, Desheng Han,
Hiroshi Hasegawa, Esa Kallio, Árpád Kis, Guan Le, Bertrand Lembège, Yu Lin, Terry Liu,
Kjellmar Oksavik, Nojan Omidi, Antonius Otto, Jie Ren, Quanqi Shi, David Sibeck, and Shutao
Yao. Dayside Transient Phenomena and Their Impact on the Magnetosphere and Ionosphere.
*Space Sci. Rev.*, 218(5):40, August 2022. doi: 10.1007/s11214-021-00865-0.
- [26] N. Omidi, J. P. Eastwood, and D. G. Sibeck. Foreshock bubbles and their global magnetospheric
impacts. *Journal of Geophysical Research (Space Physics)*, 115(A6):A06204, June 2010. doi:
10.1029/2009JA014828.
- [27] C. F. Kennel and H. E. Petschek. Limit on Stably Trapped Particle Fluxes. *J. Geophys. Res.*,
71:1–28, January 1966.
- [28] D. Summers. Quasi-linear diffusion coefficients for field-aligned electromagnetic waves with
applications to the magnetosphere. *J. Geophys. Res.*, 110:A08213, August 2005. doi:
10.1029/2005JA011159.
- [29] Terry Z. Liu, San Lu, Vassilis Angelopoulos, Heli Hietala, and Lynn B. Wilson. Fermi accel-
eration of electrons inside foreshock transient cores. *Journal of Geophysical Research (Space*
*Physics)*, 122(9):9248–9263, September 2017. doi: 10.1002/2017JA024480.
- [30] H. Hietala, A. Sandroos, and R. Vainio. Particle Acceleration in Shock-Shock Interaction:
Model to Data Comparison. *Astrophys. J. Lett.*, 751(1):L14, May 2012. doi: 10.1088/2041-
8205/751/1/L14.
- [31] J. L. Burch, T. E. Moore, R. B. Torbert, and B. L. Giles. Magnetospheric Multiscale Overview
and Science Objectives. *Space Sci. Rev.*, 199:5–21, March 2016. doi: 10.1007/s11214-015-0164-
9.
- [32] C. T. Russell, B. J. Anderson, W. Baumjohann, K. R. Bromund, D. Dearborn, D. Fischer,
G. Le, H. K. Leinweber, D. Leneman, W. Magnes, J. D. Means, M. B. Moldwin, R. Nakamura,
D. Pierce, F. Plaschke, K. M. Rowe, J. A. Slavin, R. J. Strangeway, R. Torbert, C. Hagen,
I. Jernej, A. Valavanoglou, and I. Richter. The Magnetospheric Multiscale Magnetometers.
*Space Sci. Rev.*, 199:189–256, March 2016. doi: 10.1007/s11214-014-0057-3.
- [33] O. Le Contel, A. Retinò, H. Breuillard, L. Mirioni, P. Robert, A. Chasapis, B. Lavraud,
478 T. Chust, L. Rezeau, F. D. Wilder, D. B. Graham, M. R. Argall, D. J. Gershman, P.-A.
Lindqvist, Y. V. Khotyaintsev, G. Marklund, R. E. Ergun, K. A. Goodrich, J. L. Burch, R. B.
Torbert, J. Needell, M. Chutter, D. Rau, I. Dors, C. T. Russell, W. Magnes, R. J. Strangeway,
481 K. R. Bromund, H. K. Leinweber, F. Plaschke, D. Fischer, B. J. Anderson, G. Le, T. E. Moore,
C. J. Pollock, B. L. Giles, J. C. Dorelli, L. Avakov, and Y. Saito. Whistler mode waves and

- Hall fields detected by MMS during a dayside magnetopause crossing. *Geophys. Res. Lett.*,
43:5943–5952, June 2016. doi: 10.1002/2016GL068968.
- [34] C. Pollock, T. Moore, A. Jacques, J. Burch, U. Gliese, Y. Saito, T. Omoto, L. Avanov, A. Bar-
rie, V. Coffey, J. Dorelli, D. Gershman, B. Giles, T. Rosnack, C. Salo, S. Yokota, M. Adrian,
C. Aoustin, C. Auletti, S. Aung, V. Bigio, N. Cao, M. Chandler, D. Chornay, K. Christian,
G. Clark, G. Collinson, T. Corris, A. De Los Santos, R. Devlin, T. Diaz, T. Dickerson, C. Dick-
son, A. Diekmann, F. Diggs, C. Duncan, A. Figueroa-Vinas, C. Firman, M. Freeman, N. Galassi,
490 K. Garcia, G. Goodhart, D. Guererro, J. Hageman, J. Hanley, E. Hemminger, M. Holland,
491 M. Hutchins, T. James, W. Jones, S. Kreisler, J. Kujawski, V. Lavu, J. Lobell, E. LeCompte,
492 A. Lukemire, E. MacDonald, A. Mariano, T. Mukai, K. Narayanan, Q. Nguyen, M. Onizuka,
493 W. Paterson, S. Persyn, B. Piepgrass, F. Cheney, A. Rager, T. Raghuram, A. Ramil, L. Reichen-
494 thal, H. Rodriguez, J. Rouzaud, A. Rucker, Y. Saito, M. Samara, J.-A. Sauvaud, D. Schuster,
495 M. Shappirio, K. Shelton, D. Sher, D. Smith, K. Smith, S. Smith, D. Steinfeld, R. Szymkiewicz,
496 K. Tanimoto, J. Taylor, C. Tucker, K. Tull, A. Uhl, J. Vloet, P. Walpole, S. Weidner, D. White,
G. Winkert, P.-S. Yeh, and M. Zeuch. Fast Plasma Investigation for Magnetospheric Multiscale.
*Space Sci. Rev.*, 199:331–406, March 2016. doi: 10.1007/s11214-016-0245-4.
- [35] J. B. Blake, B. H. Mauk, D. N. Baker, P. Carranza, J. H. Clemmons, J. Craft, W. R. Crain,
500 A. Crew, Y. Dotan, J. F. Fennell, R. H. Friedel, L. M. Friesen, F. Fuentes, R. Galvan, C. Ibscher,
501 A. Jaynes, N. Katz, M. Lalic, A. Y. Lin, D. M. Mabry, T. Nguyen, C. Pancratz, M. Redding,
G. D. Reeves, S. Smith, H. E. Spence, and J. Westlake. The Fly’s Eye Energetic Particle
Spectrometer (FEEPS) Sensors for the Magnetospheric Multiscale (MMS) Mission. *Space Sci.*
*Rev.*, 199:309–329, March 2016. doi: 10.1007/s11214-015-0163-x.
- [36] G. Paschmann and S. J. Schwartz. ISSI Book on Analysis Methods for Multi-Spacecraft Data.
In Robert A. Harris, editor, *Cluster-II Workshop Multiscale / Multipoint Plasma Measurements*,
volume 449 of *ESA Special Publication*, page 99, February 2000.
- [37] V. Angelopoulos. The THEMIS Mission. *Space Sci. Rev.*, 141:5–34, December 2008. doi:
10.1007/s11214-008-9336-1.
- [38] H. U. Auster, K. H. Glassmeier, W. Magnes, O. Aydogar, W. Baumjohann, D. Constantinescu,
D. Fischer, K. H. Fornacon, E. Georgescu, P. Harvey, O. Hillenmaier, R. Kroth, M. Ludlam,
Y. Narita, R. Nakamura, K. Okrafka, F. Plaschke, I. Richter, H. Schwarzl, B. Stoll, A. Vala-
vanoglou, and M. Wiedemann. The THEMIS Fluxgate Magnetometer. *Space Sci. Rev.*, 141:
235–264, December 2008. doi: 10.1007/s11214-008-9365-9.
- [39] J. P. McFadden, C. W. Carlson, D. Larson, M. Ludlam, R. Abiad, B. Elliott, P. Turin, M. Mar-
ckwordt, and V. Angelopoulos. The THEMIS ESA Plasma Instrument and In-flight Calibration.
*Space Sci. Rev.*, 141:277–302, December 2008. doi: 10.1007/s11214-008-9440-2.
- [40] Christina Chu, Hui Zhang, David Sibeck, Antonius Otto, QiuGang Zong, Nick Omid, James P.
McFadden, Dennis Fruehauff, and Vassilis Angelopoulos. THEMIS satellite observations of hot
flow anomalies at Earth’s bow shock. *Annales Geophysicae*, 35(3):443–451, March 2017. doi:
10.5194/angeo-35-443-2017.
- [41] I. Y. Vasko, V. V. Krasnoselskikh, F. S. Mozer, and A. V. Artemyev. Scattering by the broad-
band electrostatic turbulence in the space plasma. *Physics of Plasmas*, 25(7):072903, July 2018.
doi: 10.1063/1.5039687.

- [42] T. H. Stix. *The Theory of Plasma Waves*. 1962.
- [43] A. Voshchepynets, V. Krasnoselskikh, A. Artemyev, and A. Volokitin. Probabilistic model of
beam-plasma interaction in randomly inhomogeneous plasma. *The Astrophysical Journal*, 807
(1):38, 2015. URL <http://stacks.iop.org/0004-637X/807/i=1/a=38>.
- [44] Xin Tao, Anthony A. Chan, Jay M. Albert, and James A. Miller. Stochastic modeling of multi-
dimensional diffusion in the radiation belts. *Journal of Geophysical Research (Space Physics)*,
113(A7):A07212, July 2008. doi: 10.1029/2007JA012985.
- [45] Xiaofei Shi, David S. Tonoian, Anton V. Artemyev, Xiao-Jia Zhang, and Vassilis Angelopoulos.
Electron resonant interaction with whistler-mode waves around the Earth’s bow shock I: The
probabilistic approach. *Physics of Plasmas*, 30(12):122902, 12 2023. ISSN 1070-664X. doi:
10.1063/5.0172231. URL <https://doi.org/10.1063/5.0172231>.
- [46] J. G. Kirk and P. Schneider. Particle Acceleration at Shocks: A Monte Carlo Method. *Astro-
phys. J.*, 322:256, November 1987. doi: 10.1086/165720.
- [47] A. M. Bykov, D. C. Ellison, and S. M. Osipov. Nonlinear Monte Carlo model of superdiffusive
shock acceleration with magnetic field amplification. *Phys. Rev. E*, 95(3):033207, March 2017.
doi: 10.1103/PhysRevE.95.033207.
- [48] Terry Z. Liu, San Lu, Vassilis Angelopoulos, Yu Lin, and X. Y. Wang. Ion Acceleration In-
side Foreshock Transients. *Journal of Geophysical Research (Space Physics)*, 123(1):163–178,
January 2018. doi: 10.1002/2017JA024838.
- [49] O. Kruparova, V. Krupar, J. Á afránková, Z. Němeček, M. Maksimovic, O. Santolik, J. Soucek,
F. Němec, and J. Merka. Statistical Survey of the Terrestrial Bow Shock Observed by the
Cluster Spacecraft. *Journal of Geophysical Research (Space Physics)*, 124(3):1539–1547, March
2019. doi: 10.1029/2018JA026272.
- [50] A. Lalti, Yu. V. Khotyaintsev, A. P. Dimmock, A. Johlander, D. B. Graham, and V. Olshevsky.
A Database of MMS Bow Shock Crossings Compiled Using Machine Learning. *Journal of Geo-
physical Research (Space Physics)*, 127(8):e30454, August 2022. doi: 10.1029/2022JA030454.

**Acknowledgments**

We are thankful to the THEMIS and MMS teams and instrument principal investigators for excellent
data making possible this study.

Some of the work was supported by the Geospace Environment Modeling (GEM) Focus Group
entitled, "Particle Heating and Thermalization in Collisionless Shocks in the Magnetospheric Mul-
tiscule Mission (MMS) Era," led by L.B. Wilson III.

We thank the contribution of Emmanuel Masongsong for helping with the first illustration figure.

**Author Contributions**

X.S. contributed to the conceptualization of the study, data analysis, data interpretation, numerical
simulations, and the manuscript's writing. A.A. conducted the theoretical model. V.A., T.L., and
561 L.B.W. contributed to analyzing observations and simulations data, and to data interpretation. All
562 authors participated in the manuscript reviewing and editing.

**Competing Interests**

The authors declare no competing interests.

Response to reviewers' comments on *Compound electron acceleration at planetary foreshocks*

We are grateful to the reviewers's constructive responses. We also apologize that some of our changes in response to previous reviews have not been appropriately explained and incorporated into the text. We have significantly extended our Method section to address the reviewer's concerns. Our point-by-point responses to the comments are listed below in blue and the track changes file is also attached.

Response to the Reviewer # 1

I have read the authors' response and the revised manuscript by Shi et al. that is under consideration in Nature Communications. I found that the authors' revision was not satisfactory, and consequently, I do not recommend it for publication. My opinion is based on the following two reasons: (1) the authors did not address some of the concerns raised in my previous report, and (2) the authors' model may contain physically unreasonable assumptions, which I found difficult to assess from the description given in the paper. The paper would be an interesting contribution to a more specialized journal.

First, we should apologize if some of Reviewers' questions have not addressed properly. We did not intent to ignore or reduce significance of any questions. Thanks to Editor, we have a second chance to make a necessary modifications of the model description, and we plan to do our best to clarify all model elements. We also should note from the beginning that 1D nature of the acceleration model is definitely strong simplification that does not allow us to incorporate such 2D effects as magnetic field line divergence/convergence around the foreshock transients and charged particle escape along flanks of the acceleration region. This simplification is necessary because our model is based on incorporation of wave-particle interaction effects, and thus require certain prescribed "unperturbed" particle dynamics. Below in our response and in the Method section of the main text we provide arguments and additional clarifications supporting the idea that such simplification may still keep the main model result valid.

- *The authors's description of the model is not satisfactory. They did*

not define the energy gain by SDA ΔE_{SDA} in the Method section (while they did in the response). Also, the relations between $\Delta E_{\parallel, Fermi}$ and ΔE_{Fermi} , $\Delta E_{\perp, beta}$ and ΔE_{Beta} are unclear.

We apologize for these inconsistencies and incomplete information. $\Delta E_{\parallel, Fermi}$ and ΔE_{Fermi} are the same, and $\Delta E_{\perp, beta}$ and ΔE_{Beta} are also the same. We have revised the Method section significantly, and also the main text, included all details and clarifications regarding acceleration mechanism involved. We also have included a schematic showing all elements of the model.

- *The crucial problem is that they did not describe the problem in sufficient detail. If the model considers bouncing particles between the shock and foreshock transients, Fermi acceleration (if this considers a particle bouncing between the two magnetic walls) should already contain SDA because (classical) SDA is nothing more than a mirror reflection. Similarly, what the authors meant by the betatron acceleration is also unclear. Again, if the particles are bouncing, the magnetic field strength as seen by the particles is essentially unchanged because they are in the (nearly undisturbed) solar wind.*

The Reviewer is right, the Fermi acceleration is identical to SDA and when we speak about Fermi acceleration in our model we mean that SDA effect is accounted for both bow shock and shock of the foreshock transient (two magnetic walls). The motion of the shock of the foreshock transient with velocity v_f results in the same SDA of electrons, and in the paper this acceleration is referred to as Fermi acceleration, $\Delta E_{\parallel, Fermi} \propto v_f^2$, to separate it from the SDA at the bow shock. There is no additional Fermi acceleration mechanism beyond the SDA included in our model. We have clarified this aspect in the text. Regarding the betatron acceleration: indeed, just bouncing between magnetic walls would not increase the magnetic field observed by particles and should not lead to any acceleration. In our model we take into account observational and simulation results showing that the magnetic field strength in the core region (between the bow shock and foreshock transient) can increase by factor $\sim \times 3$ when the foreshock transient approaches the bow shock and compresses the core plasma (Liu et al. [2019]). This increase of the magnetic field magnitude does not affect the Fermi/SDA acceleration, but can provide additional betatron heating. We have clarified these aspects about Fermi and betatron accelerations in the Method section (the same schematic is included into Method).

- *The authors mentioned in the response (but again not in the manuscript)*

Figure S1: A schematic view of the main elements of the electron acceleration model. Electrons are bouncing between two magnetic walls (mirrors) formed by the bow shock and the foreshock transient boundary (black curve shows the magnetic strength profile shaping the effective potential for electron trapping). With bow shock normal velocity v_{bs} , electrons experience SDA with the energy gain $\Delta E_{\parallel, \text{SDA}} \propto v_{bs}^2$. The motion of the foreshock transient boundary with velocity v_f also results in SDA acceleration of electrons, which is termed Fermi acceleration in the paper, $\Delta E_{\parallel, \text{Fermi}} \propto v_f^2$, to separate it from the SDA at the bow shock. When foreshock transient approaches the bow shock, it compress plasma in the core region (between the bow shock and foreshock transient), increasing the magnetic field magnitude (blue curve shows the magnetic strength profile with time evolution). Such increase of the magnetic field magnitude results in betatron electron acceleration, $\Delta E_{\perp, \text{beta}} \propto \delta B/B$. When electrons reaching the bow shock and the foreshock transient boundary, they experience scattering due to resonant interactions with electrostatic turbulence, high frequency whistler-mode waves, and magnetosonic waves. Additionally, high frequency whistler waves can also change electron energies during the interaction.

that the accelerated particles may spend more time in regions of compressed magnetic field. This makes the interpretation of the model even more tricky. Remember that SDA is an adiabatic process. During the interaction with the shock, particles will gain energy in the perpendicular direction due to the conservation of the adiabatic moment. In other words, their instantaneous energy gain is due to the betatron. The perpendicular energy is converted to the parallel energy through the reflection. It is hard to understand whether separating SDA and the betatron into two different mechanisms is reasonable.

We apologise for this inaccurate description. We did not include any additional effects of electrons spending more time within the foreshock transient, and we used this comment only to address the question about role of 2D effects (magnetic field line converging around the foreshock transient). The Reviewer understanding of our model as a 1D electron bouncing between approaching magnetic walls is totally correct. We also should agree that the separation of betatron and SDA does not be reasonable if we deal only with electron interaction with the bow shock and foreshock transient. We do not use the betatron to explain SDA, and betatron heating mentioned in the paper does not refer to the perpendicular energy gain during the SDA. We use it in the context of the magnetic field increase in the core (the region between the bow shock and the foreshock transient). To clarify this point we have revised the text in the Method section and added the schematic figure.

- *I might have misunderstood something. The problem in the manuscript is, however, that the description given in the paper is too vague, and there is no way for the readers to judge the validity. Although it might be a reasonable model under some specific conditions, I believe that the information given in the manuscript is far from sufficient.*

We have to agree with the Reviewer, the previous version of the Method section may be confusing in details of acceleration and scattering mechanisms included into our model. We have significantly rewritten the Method section to address this issue.

- *One final rather minor comment. The authors did not provide a detailed description of the analyses (such as time intervals used) from which they estimated the parameters of the shock. This point is not necessarily critical, but as I mentioned in the previous report, the shock appears to be rather quasi-parallel rather than quasi-perpendicular. Such information should be necessary for reproducibility.*

We apologise for missed information. We have included details into the Method section.

Response to the Reviewer # 2

Thanks for addressing my points. The manuscript has improved and it is certainly interesting, though in my view, the element of novelty justifying publication in Nature Communications is still a bit weak. I have no further comments.

We are thankful to the Reviewer for positive feedback!

Response to the Reviewer # 3

In the revised version of the Manuscript authors addressed satisfactorily all the major points raised. The methods section now contains additional and detailed information on the data analysis and on the numerical model that give strength to the work. I believe that the Manuscript can be published in Nature Communications in the present form, since it presents a work that significantly advances our knowledge on electron acceleration at collisionless shocks.

We are thankful to the Reviewer for positive feedback!

References

Terry Z. Liu, Vassilis Angelopoulos, and San Lu. Relativistic electrons generated at Earth's quasi-parallel bow shock. *Science Advances*, 5(7):eaaw1368, July 2019. doi: 10.1126/sciadv.aaw1368.

Response to reviewers' comments on *Compound electron acceleration at planetary foreshocks*

Response to the Reviewer

The authors have significantly improved the quality of the manuscript. Now, at least, the model assumption itself becomes much clearer. I still have a concern about whether the quasi-perpendicular geometry employed by the authors is appropriate or not. However, the idea itself is sufficiently novel, and it would be reasonable to publish the paper for further scrutiny by the community.

We are thankful to the Reviewer for positive feedback!

- *Just a minor comment. I believe that both the coplanarity and mass-flux conservation methods require the upstream and downstream intervals. So the authors should state the two time intervals in Methods section.*

Yes, the reviewer is right that the coplanarity and mass-flux conservation methods require the two intervals. We have included this information in the Method in Lines 201-202.